# Improving deep learning protein monomer and complex structure prediction using DeepMSA2 with huge metagenomics data

Wei Zheng [1], Qiqige Wuyun [2], Yang Li [1,3], Chengxin Zhang [1], Lydia Freddolino [1,4] ✉ & Yang Zhang [1,3,4,5,6] ✉

Leveraging iterative alignment search through genomic and metagenome sequence databases, we report the DeepMSA2 pipeline for uniform protein single- and multichain multiple-sequence alignment (MSA) construction. Large-scale benchmarks show that DeepMSA2 MSAs can remarkably increase the accuracy of protein tertiary and quaternary structure predictions compared with current state-of-the-art methods. An integrated pipeline with DeepMSA2 participated in the most recent CASP15 experiment and created complex structural models with considerably higher quality than the AlphaFold2-Multimer server (v.2.2.0). Detailed data analyses show that the major advantage of DeepMSA2 lies in its balanced alignment search and effective model selection, and in the power of integrating huge metagenomics databases. These results demonstrate a new avenue to improve deep learning protein structure prediction through advanced MSA construction and provide additional evidence that optimization of input information to deep learning-based structure prediction methods must be considered with as much care as the design of the predictor itself.

Substantial progress in protein structure prediction has been witnessed in the recent community-wide Critical Assessment of protein Structure Prediction (CASP) experiments[1]. In CASP14, for example, the end-to-end deep learning protocol AlphaFold2[2] was able to create atomic-level structure predictions for two-thirds of the single-domain protein sequences[3]. AlphaFold2 was later extended to AlphaFold2-Multimer[4] for multichain protein complex structure prediction, and proved capable of generating high-quality complex models in many cases. The AlphaFold2 framework makes use of coevolutionary features derived from MSAs as the major input for self-attention networks to train and create three-dimensional (3D) protein models. The quality of the input MSAs is therefore a key factor in determining whether a high-accuracy model can be produced.

Because of the importance of MSAs for structure prediction, the development of methods to accurately detect and align a diverse set of homologous sequences represents an important direction to improve predictive accuracy[5]. The exponentially increasing size of the metagenome sequence databases has made the task of quick and accurate MSA construction highly nontrivial[6]. One recent example along this line is ColabFold[7], which aimed to accelerate the MSA generation pipeline of AlphaFold2 by replacing the MSA search program HHblits[8] with a more sensitive and faster tool, MMseqs2[9], meanwhile using nonredundant metagenomic databases. A more ambitious effort to enhance the contribution of sequence information in structure predictions is the use of a protein language model[10], which utilizes a deep learning transformer network for learning coevolutionary information by masking parts

[1]Department of Computational Medicine and Bioinformatics, University of Michigan, Ann Arbor, MI, USA. [2]Department of Computer Science and Engineering, Michigan State University, East Lansing, MI, USA. [3]Cancer Science Institute of Singapore, National University of Singapore, Singapore, Singapore. [4]Department of Biological Chemistry, University of Michigan, Ann Arbor, MI, USA. [5]Department of Computer Science, School of Computing, National University of Singapore, Singapore, Singapore. [6]Department of Biochemistry, Yong Loo Lin School of Medicine, National University of Singapore, Singapore, Singapore. ✉e-mail: lydsf@umich.edu; zhang@zhanggroup.org

of the input sequences and trains the network to recover them. Once trained, the language model can be used to perform structure predictions even without an MSA. Protein structure prediction methods that combine language models with the AlphaFold2 structure module, such as ESMFold[11] and OmegaFold[12], can generate better models than Alpha-Fold2 on some orphan sequences for which detectable homologous sequences do not exist.

Despite ongoing efforts in the field, the above methods do not substantially improve the overall prediction accuracy for monomer proteins. In addition, structure prediction for protein complexes remains an even more substantial challenge. In the CASP14 experiment, for example, satisfactory models (those with an Interface Contact Score (ICS) >0.8) could be built for only 7% of tested protein complexes[13]. The situation was considerably improved in CASP15, where the best performance methods (including the pipeline introduced in this study) provided satisfactory models for up to 47% of cases[14]. However, there is still no evidence that modifications made in the newly introduced AlphaFold2-based methods (for example, ColabFold) can obviously improve the performance of protein complex modeling relative to AlphaFold2. It is also notable that in the CASP15 experiment, ostensibly MSA-free language model-based methods such as OmegaFold, performed poorly on targets with few homologous sequences[15], suggesting that the lack of sufficient evolutionary information encoded in the protein language models is equivalently problematic to shallow MSAs for explicitly MSA-based methods.

To systematically explore the potential contributions of optimal MSAs for protein structure prediction, we present DeepMSA2 (Fig. 1), a hierarchical approach inspired by our previous iterative monomer MSA construction method, DeepMSA[5]. Compared with DeepMSA, in addition to the protocol extension from monomers to multimers, DeepMSA2 couples several newly developed MSA generation pipelines to create multiple MSAs based on huge genomics and metagenomics sequence databases containing a total of 40 billion sequences and introduces a deep learning-driven MSA scoring strategy for optimal MSA selection. We present careful benchmarks for DeepMSA2 applied to large-scale datasets containing both monomer and multimer targets from recent CASP13–15 experiments, with results demonstrating substantial advantages of the pipeline for improving both protein tertiary and quaternary structure modeling accuracy compared with contemporary state-of-the-art approaches. We have made DeepMSA2 and associated structural databases freely available to the community, and the results of this study should have important implications for future developments of new MSA construction and deep learning protein structure and function prediction methods.

## Results

DeepMSA2 consists of two separate pipelines for monomer and multimer MSA construction, respectively. For monomer MSA construction (Fig. 1a), it utilizes three parallel blocks (dMSA, quadrupole MSA (qMSA) and mMSA) built on different searching strategies to obtain raw MSAs from a diverse set of databases, assembled from whole-genome and metagenome sequence libraries. In each of the three MSA generation blocks, a similar logic is followed, in which an initial query is searched against a sequence database, and if a sufficient number of effective sequences is not achieved, iterative searches into larger databases are attempted. Up to ten raw MSAs gathered from the three blocks are ranked through a rapid deep learning-guided prediction process to select the optimal MSA. For multimeric MSA construction (Fig. 1b), multiple composite sequences are created by linking monomeric sequences from different component chains that have the same orthologous origins. Here, a set of $M$ top-ranked monomeric MSAs from each chain are paired with those of other chains, which results in $M^N$ hybrid multimeric MSAs with $N$ being the number of distinct monomer chains in the complex. The optimal multimer MSAs are then selected based on a combined score of the depth of the MSAs and folding score (predicted

local Distance Difference Test (pLDDT) score) of the monomer chains as defined in equation (3) below. Full details for both MSA construction procedures are provided in Methods.

### Improvements of monomer structure prediction by DeepMSA2

We first tested the performance of template recognition and deep learning-based spatial restraint prediction assisted by the monomer MSAs produced by DeepMSA2 and five other commonly used pipelines: BLAST[16], HHblits[8], HMMER[17], MMseqs2[9] and PSIBLAST[18]. Overall, DeepMSA2 performs better than the five control programs in all three assessment criteria, including average template modeling scores (TM-scores) of the structure templates recognized by HHsearch (Fig. 2a and Supplementary Table 1), the precision of the top $L$ long-range contacts predicted by DeepPotential[19] ($L$ is the sequence length and 'long-range' represents the sequence separation $|i − j| \geq 24$) (Fig. 2b and Supplementary Table 2), and the mean absolute distance error (MAE; equation (6)) of the top $5L$ long-range distances (Fig. 2c, Supplementary Fig. 1 and Supplementary Table 3). The number of effective sequences (Neff; equation (1)), the average sequence identity and alignment coverage are also compared between those six MSAs (Supplementary Fig. 2 and Supplementary Table 4), where DeepMSA2 shows the ability in collecting homologous sequences with more balanced alignment coverage and diversity. A detailed discussion on the features of the monomeric MSAs and the potential impact on template recognition and spatial restraint prediction is summarized in Supplementary Discussion Text 1.

As a more direct test of DeepMSA2 on deep learning 3D structure prediction, we implement a modified version of AlphaFold2[20], in which the input MSA is replaced with the MSA created by DeepMSA2. For brevity, we use 'DeepMSA2-based protein folding' (DMFold) to refer to this hybrid pipeline in the following discussion. In Fig. 3a, we compare the TM-scores of all models predicted by DMFold versus AlphaFold2 on the 132 free modeling (FM) monomer proteins from the CASP13–15 experiments. To correctly reflect the FM nature of the domains, all templates released after May 2018, May 2020 and May 2022 have been excluded for the CASP13, CASP14 and CASP15 domains, respectively, when running the programs. It is shown that DMFold generated models with a higher TM-score than AlphaFold2 in 63% (83 of 132) of cases. The average TM-score of the models generated by DMFold (0.821) is 5% higher than that generated by AlphaFold2 (0.781), with a $P$ value of $1.82 \times 10^{-4}$ in a one-sided Student's $t$-test indicating that the difference is statistically significant. It is notable that the difference mainly comes from difficult domains. For the 86 domains where both AlphaFold2 and DMFold achieved a TM-score >0.8, for example, the average TM-score is very close (0.925 for DMFold versus 0.922 for AlphaFold2). However, for the remaining 46 domains, where at least one of the methods performed poorly, the difference in TM-score is dramatic (0.626 for DMFold versus 0.517 for AlphaFold2; $P = 2.86 \times 10^{-4}$, one-sided Student's $t$-test). Among the 46 difficult domains, DMFold builds models with TM-scores 0.1 unit higher than AlphaFold in 18 domains, whereas AlphaFold2 does so only in 4 domains.

In Supplementary Table 5 we further list the statistics of sequences in the MSAs by AlphaFold2 and DMFold. Although AlphaFold2 collects a slightly larger number of homologous sequences than DeepMSA2 (2,724 versus 2,279), the average number of effective sequences in the DeepMSA2 MSAs (Neff = 93.7) is much higher than that for the AlphaFold2 MSAs (Neff = 84.5), suggesting that DeepMSA2 manages to identify more diverse homologous sequences and build a 'deeper' effective MSA than AlphaFold2's default MSA pipeline. One reason for this improvement is that the inclusion of in-house metagenome sequences derived from the Tara database (TaraDB)[21], MetaSource database (MetaSourceDB)[6] and JGIclust database substantially increased the coverage of biological sequence space. In addition, the multilevel iterative searching performed by DeepMSA2 helps it to collect more diverse but

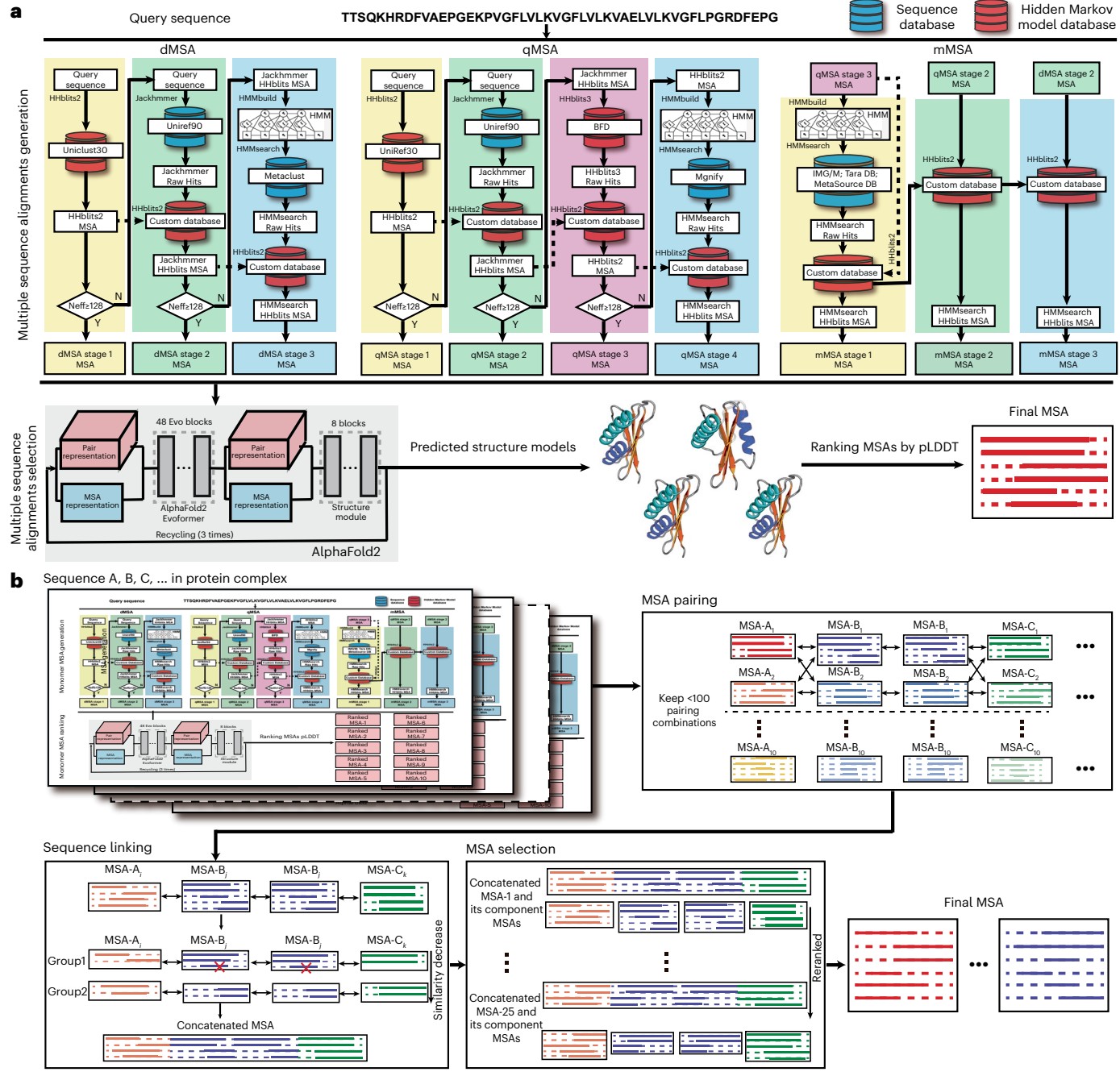

**Fig. 1 | Schematic of the DeepMSA2 pipelines for monomeric and multimeric MSA constructions. a**, DeepMSA2-Monomer contains two steps: an iterative MSA generation step that combines the dMSA, qMSA and mMSA algorithms, and a deep learning-based MSA ranking step based on the confidence scores of predicted structure models. **b**, DeepMSA2-Multimer contains four steps of monomeric MSA generation, MSA pairing, sequence linking and concatenated MSA selection. IMG/M is the metagenomics database sourced from Joint Genome Institute, see JGIclust in Methods.

relevant homologous sequences. In the bottom row of Supplementary Table 5, we list the modeling results obtained by using DMFold without the in-house metagenome sequence databases (referred to 'DMFold-noh'). It shows that DMFold-noh still outperforms AlphaFold2 in both Neff and TM-score. However, DMFold-noh clearly underperforms the full version DMFold, with a significantly lower TM-score ($P = 1.65 \times 10^{-5}$, one-sided Student's *t*-test). This suggests that both the enhanced sequence databases and searching algorithms in DeepMSA2 contribute to the quality of MSAs and structural model construction.

In Fig. 3b, we present a case study of the FM domain T1043-D1 from CASP14, for which AlphaFold2 generates an incorrect model with a TM-score of 0.20 and a global pLDDT of 0.40. Here, pLDDT is a scale

used by AlphaFold2 to evaluate the residue-level prediction quality, with pLDDT ≥ 0.7 indicating a correct backbone fold, and pLDDT < 0.7 indicating an expected failure to fold the protein[20]. The poor result observed here is mainly due to insufficient coevolutionary information, because the AlphaFold2 default MSA pipeline detects only two homologous sequences, resulting in a Neff value of 0.16 in the MSA. Figure 3c shows the number of aligned amino acids per residue ($N_r$) and the pLDDT score along the protein sequence. Overall, the residue-level pLDDT scores show a strong correlation with $N_r$, demonstrating again the importance of MSA information in driving structure prediction. By contrast, DMFold constructs a model with a TM-score of 0.73 and a pLDDT of 0.71. The improvement in modeling quality by DMFold is

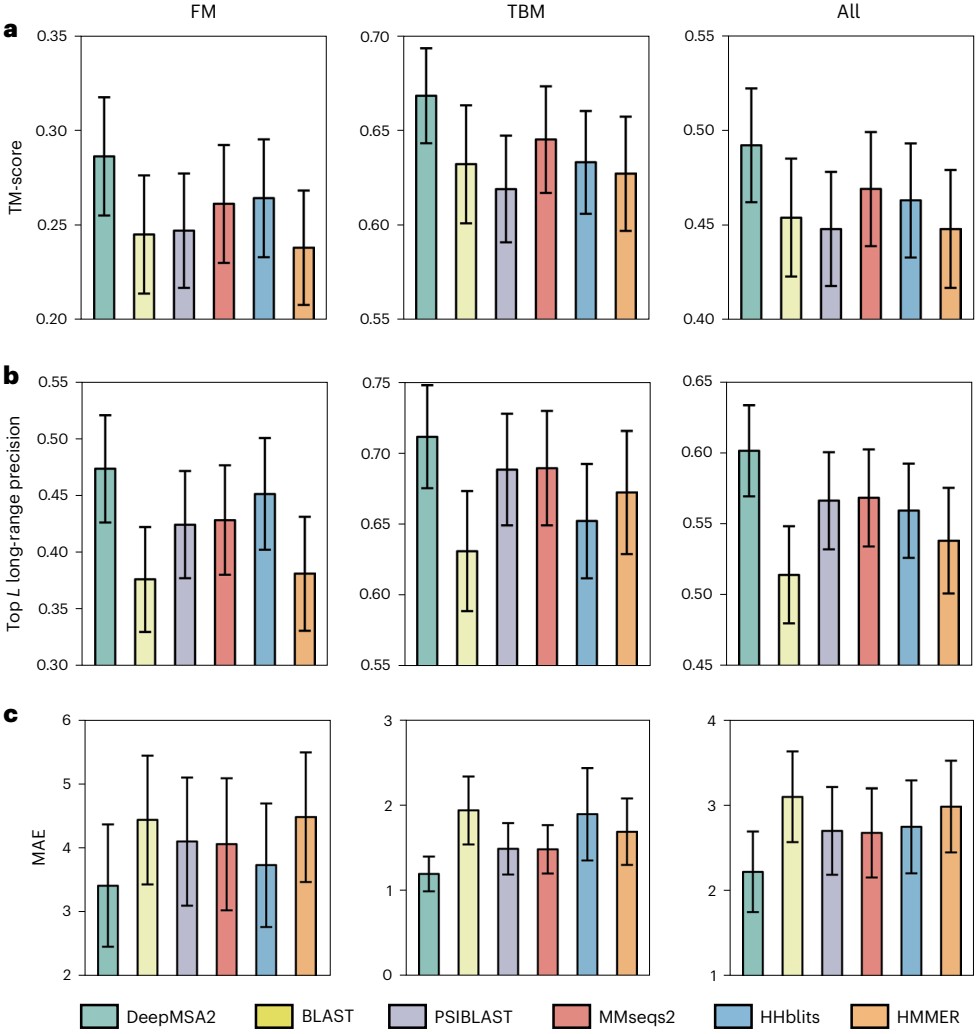

**Fig. 2 | Comparisons of MSAs generated by DeepMSA2 and five control methods for assisting template recognition and deep learning spatial restraint prediction on 293 CASP13–15 monomer domains. a**, Average TM-score of the first template detected by HHsearch. **b**, Precision of top $L$ long-range residue–residue contact prediction with $L$ being the sequence length and sequence separation $|i − j| \geq 24$. **c**, MAE for the top $5L$ long-range residue–residue distance predictions by DeepPotential. The height of the histogram indicates the mean value and the error bar depicts the 95% confidence interval for each variable using Student's $t$-distribution. The CASP domains are categorized into FM and TBM by the accessors. In **a**, $n$ = 287, 155 and 132 monomer domains for the columns 'All', 'TBM' and 'FM' respectively, whereas in **b** and **c**, $n$ = 271, 146 and 125 for the three columns, respectively.

mainly because DeepMSA2 constructs a deeper MSA with 42 homologous sequences and a Neff value of 2.2, which offers more helpful coevolutionary information. The difference is especially dramatic in the N-terminal portion of the protein, where the DeepMSA2 alignments have an especially high $N_r$, and the pLDDT scores increase accordingly (Fig. 3d). In Supplementary Fig. 4, we list eight other examples from CASP13–15 (T0991-D1, T1064-D1, T1125-D1, T1125-D2, T1125-D5, T1130-D1, T1169-D1 and T1169-D4), in which the TM-score improvements by DMFold are >0.3. In seven of these eight cases, the Neff of DeepMSA2 is higher than that of AlphaFold2. These results again highlight the capacity of DeepMSA2 to provide more informative MSAs to a state-of-the-art protein prediction pipeline, thus further improving protein monomer modeling accuracy and rendering many previously 'unfoldable' proteins tractable for structure prediction.

### Human proteome modeling for difficult proteins with DeepMSA2

To further examine the practical usefulness of our new developments for large-scale structure modeling, we applied the DeepMSA2/DMFold pipeline to the human proteome. Considering the availability of the AlphaFold2 Structure Database (DB), which was recently released by the DeepMind team[22], our focus is on the 5,042 difficult sequences for which the AlphaFold2 DB models have a confidence score of pLDDT < 0.7. In Fig. 4a, we show the histogram distributions of fold-level pLDDTs obtained by DMFold and AlphaFold2 DB on these difficult proteins, where a clear shift is observed for the DMFold models towards higher pLDDT values. On average, the pLDDT of DMFold models (0.663) is 11% higher than that of AlphaFold2 DB models ($P < 2.2 \times 10^{-16}$, one-sided Student's $t$-test), and 94% (4,738 of 5,042) of the DMFold models had a higher pLDDT than the corresponding AlphaFold2 DB model. Overall, DMFold creates high-quality global folds with pLDDT ≥ 0.7 for 1,934 proteins that AlphaFold2 failed to model.

In Supplementary Fig. 5, we plot a histogram distribution of TM-scores between DMFold and AlphaFold2 DB models for the 1,934 proteins that could be folded only by DMFold. Eighty percent (1,549 of 1,934) of the DMFold models have a different overall structure relative to the corresponding AlphaFold2 DB models (with a TM-score of <0.6 between them), indicating that the improvement in the DMFold models is at the topology level. By contrast, for the remaining 385 proteins, DMFold models have relatively similar structures to AlphaFold2 DB

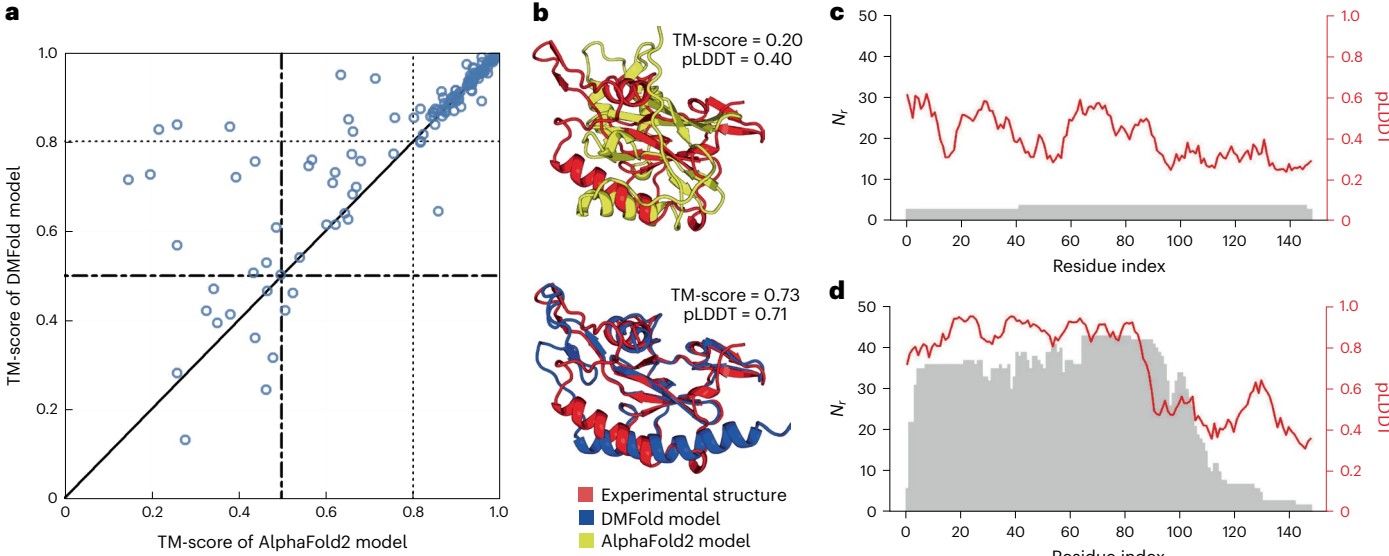

**Fig. 3 | Benchmark studies of DMFold predictions on protein monomer domains compared with AlphaFold2. a**, Head-to-head comparison of TM-scores between DMFold and AlphaFold2 on the 132 FM monomer domains from CASP13, CASP14 and CASP15. **b**–**d**, Case study for an FM domain T1043-D1, displaying the structural models by AlphaFold2 and DMFold (**b**), and the number of aligned residues per site ($N_r$; gray) and pLDDT score (red) along the protein sequence using the MSAs by AlphaFold2 (**c**) and DeepMSA2 (**d**), respectively.

models, and thus the improvements in DMFold may come mainly from local structural corrections. Figure 4b further shows a head-to-head comparison of the residue-level pLDDTs obtained by DMFold and AlphaFold2 DB for the 1,934 human proteins, which involve a total of 878,094 residues, where DMFold models have a higher residue-level pLDDT than the corresponding AlphaFold2 DB models on 93% of the residues.

In Fig. 4c, we present one illustrative example from an uncharacterized protein, Q6ZQT0, for which AlphaFold2 collects only nine homologous sequences with Neff = 0.7 in the MSA, compared with the DeepMSA2 MSA with 122 sequences and Neff = 6.2. Because of the sparse information from the MSA, a poor structure model with pLDDT = 0.51 is produced by AlphaFold2, showing an irregular secondary structure. By contrast, using the improved MSA from DeepMSA2, DMFold creates a model with much higher confidence (pLDDT = 0.92), which has a more stable fold with a well-formed hydrogen-bonding network and secondary structure. Figure 4d further lists the residue-level pLDDT distributions, where nearly all residues in the DMFold model have a pLDDT of >0.7; the corresponding residues in the AlphaFold2 DB model all fell below 0.7. For this protein, the DMFold model and AlphaFold2 DB model have a very low similarity with a TM-score of 0.44, showing that DMFold improves the quality of the global fold. Figure 4e shows a complementary example from the putative diacylglycerol O-acyltransferase 2-like protein (Q6IED9) with an α/β three-layer sandwich fold. Although the DMFold and AlphaFold2 models have similar global folds (TM-score = 0.88), DMFold built the model with a pLDDT of 0.83, whereas the AlphaFold2 DB model has a pLDDT of 0.68. The residue-level pLDDT distributions in Fig. 4f show that DMFold created better local structures with greater pLDDTs for several regions (marked in red), corresponding to two better-formed β-sheets in the 3D structural packing, as highlighted in red in Fig. 4e. These examples show that DMFold could improve AlphaFold2 modeling at both the global fold and local structure levels by supplying additional evolutionary information from more informative MSAs.

Of the 5,042 human proteins for which no high-confidence AlphaFold2 structure was available, 48 have experimental structures that cover >80% of the sequence of the natural protein and were released in the PDB after the model training date of AlphaFold2 (1 May 2018). For these 48 proteins, AlphaFold2 DB models achieve an average TM-score of 0.630, compared with 0.679 for DMFold ($P = 1.46 \times 10^{-4}$, one-sided Student's t-test; Supplementary Table 6). Supplementary Fig. 6 examines the correlation between the TM-score and pLDDT of DMFold for those 48 proteins. Among all models with a DMFold pLDDT ≥0.7, 85% of the predictions could be considered as true positives; that is, the model is predicted as foldable and is actually foldable with a TM-score >0.5. There is also a quite high false omission rate (76%) based on the 0.7 pLDDT score cutoff, suggesting that many of the models with a lower pLDDT might also possess correct folds. Overall, we note that despite the promising results for the small set of recently crystallized human proteins, the absolute quality of the predicted human proteome models from DMFold should be further verified with more proteins when the experimentally solved structures are available in the future.

**Improvements of protein complex structure prediction**

To examine the impact of DeepMSA2-Multimer on protein complex structure modeling, we collected 54 complex targets from CASP13 and CASP14 each of which contains between two and eight chains; 40 of the targets are homomers and 14 are heteromeric complexes (Supplementary Table 7). In Supplementary Table 8, we list a summary of TM-score comparisons for the complex models constructed by AlphaFold2-Multimer and DMFold-Multimer, which replaces the default MSA of AlphaFold2-Multimer by the multimer MSA from DeepMSA2-Multimer (Methods). It is found that DMFold-Multimer generates models for all, heteromer and homomer complexes with TM-scores of 0.834, 0.930 and 0.801, which are 12.2%, 3.9% and 16.1% higher than those of AlphaFold2-Multimer models (0.743, 0.895 and 0.690), respectively. The P value in a one-sided Student's t-test is below 0.05 in all the comparisons, indicating that all the differences are statistically significant. Figure 5a also shows a head-to-head comparison of the TM-score of the models, where DMFold-Multimer outperforms AlphaFold2-Multimer in 70% of cases. Again, the improvement mainly occurs on the difficult complexes. If we consider the 26 easy targets for which both DMFold-Multimer and AlphaFold2-Multimer models have TM-scores >0.9, the average TM-scores are very close (0.961 for DMFold-Multimer versus 0.960 for AlphaFold2-Multimer). For the 28 more difficult targets, however, the average TM-score of DMFold-Multimer (0.716) is significantly higher than that of

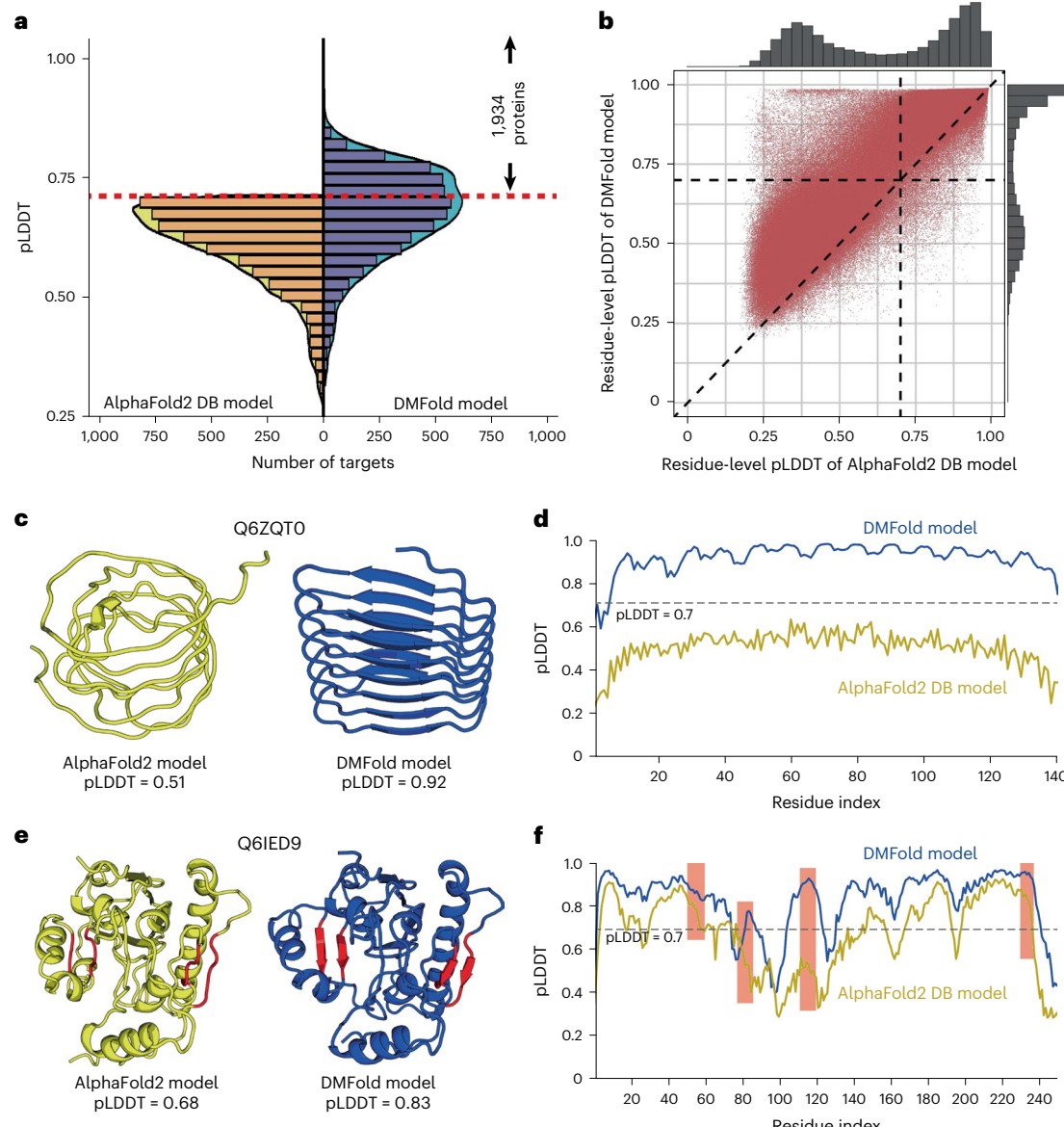

**Fig. 4 | The structural modeling results of DMFold on 5,042 difficult targets from the human proteome. a**, Distributions of pLDDTs for DMFold models versus AlphaFold2 DB models for the subset of 5,042 AlphaFold2 DB models with pLDDT < 0.70. The red dashed line marks the threshold pLDDT = 0.70 for considering a target to be confidently predicted, where DMFold models have a pLDDT ≥ 0.7 in 1,934 cases. **b**, Head-to-head comparison of the residue-level pLDDTs obtained by DMFold and AlphaFold2 DB for the 1,934 confidently DMFold-modeled proteins, which involve in total 878,094 residues. **c**, Structural models generated for a putative uncharacterized protein FLJ45035 (Q6ZQT0) by

AlphaFold2 DB (yellow) and DMFold (blue), respectively. **d**, Residue-level pLDDT curves of AlphaFold2 DB (yellow) and DMFold (blue) for Q6ZQT0. **e**, Structural models generated for a putative diacylglycerol *O*-acyltransferase 2-like protein (Q6IED9) by AlphaFold2 DB (yellow) and DMFold (blue), respectively, where two better-formed β-sheet secondary structures created by DMFold are highlighted by red. **f**, Residue-level pLDDT curves of AlphaFold2 DB (yellow) and DMFold (blue) for Q6IED9, where the pLDDTs associated with the four β-strands are highlighted with red backgrounds.

AlphaFold2-Multimer (0.542), with $P = 1.05 \times 10^{-4}$ in a one-sided Student's *t*-test.

Compared with the default MSAs in AlphaFold2-Multimer, two factors may contribute to the quality improvement of Deep-MSA2-Multimer MSAs. One is the integrated MSA creation, pairing and selection mechanism of DeepMSA2-Multimer, and the second is the inclusion of the additional huge in-house metagenomics databases. To assess the relative contributions of these factors, in Fig. 5b we compare the complex modeling performance of AlphaFold2-Multimer and DMFold-Multimer using different sequence databases. Even with the same sequence databases (from genomic sequences, Big Fantastic Database (BFD) and Mgnify), DMFold-Multimer still outperforms AlphaFold2-Multimer with the TM-score increasing from 0.743

to 0.784, indicating the usefulness of DeepMSA2-Multimer's MSA generating, pairing and selection methods. After using the full version DMFold-Multimer including our expanded metagenome databases, the modeling quality can be further increased by 6.4% from 0.784 to 0.834, showing that the large metagenome databases are also beneficial for protein complex modeling.

Figure 5b also indicates that the magnitude of TM-score improvement of DMFold-Multimer over AlphaFold2-Multimer is relatively small for heteromers compared with that for homomer complexes. This is probably because of the sequence linking mechanism, in which DeepMSA2-Multimer links two sequences if they come from the same species based on the UniProt species annotation to ensure orthologous pairing of the protein interactions. Because of the limit of species

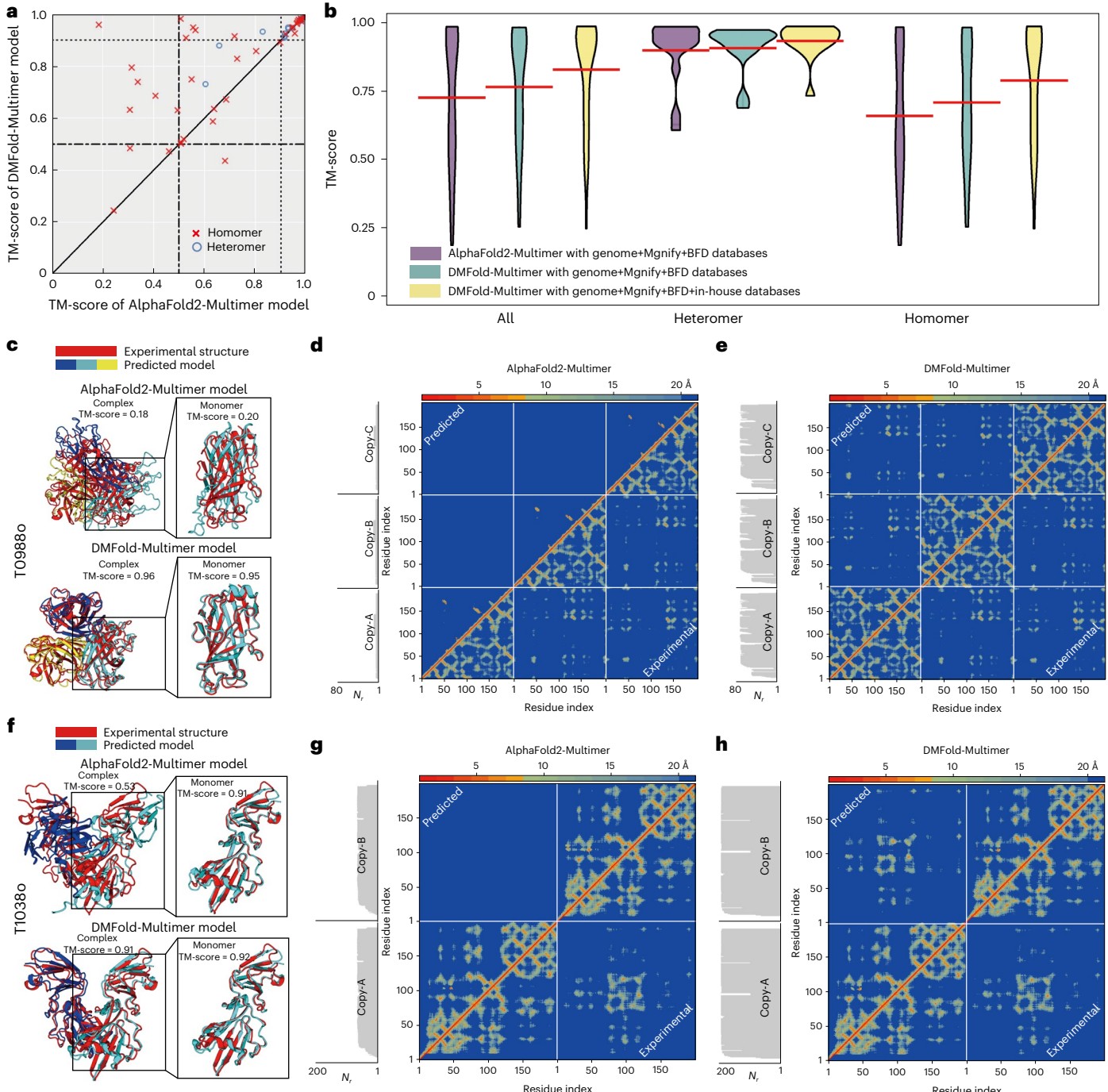

**Fig. 5 | Benchmark results of DMFold-Multimer on protein complex structure prediction. a**, Comparison of TM-scores between DMFold-Multimer and AlphaFold2-Multimer on 14 heteromer and 40 homomer complexes from CASP13 and CASP14. **b**, Violin plot of TM-scores for the targets from **a** using DMFold-Multimer (with different databases) and AlphaFold2-Multimer. Red lines show the mean of each distribution. **c**, AlphaFold2-Multimer and DMFold-Multimer models for T0988o superposed with the experimental structure, where monomers of the predicted models are colored differently. **d,e**, Residue–residue distance map (heat map) along with the number of aligned residues per site ($N_r$, shown in margins) predicted from AlphaFold2-Multimer (**d**) and DMFold-Multimer (**e**) (above red diagonal line) versus that calculated from the experimental structure (below red diagonal line) for T0988o. **f**, AlphaFold2-Multimer and DMFold-Multimer models for T1038o superposed with the experimental structure. **g,h**, Residue–residue distance map (heat map) along with the number of aligned residues per site ($N_r$, shown in margins) predicted from AlphaFold2-Multimer (**g**) and DMFold-Multimer (**h**) (above red diagonal line) versus that calculated from the experimental structure (below red diagonal line) for for T1038o.

annotations, only homologous sequences from genomic databases in the MSAs of the individual chains can be used for linking, as sequences from metagenomics databases do not have species annotations from UniProt. Thus, one major advantage of DeepMSA2, which leverages information from large metagenomics databases, will be eliminated because of the absence of species annotations (although the improved

MSAs from monomer MSA generation and complex sequence pairing and selection still contribute to more accurate structure prediction). By contrast, for homomer complexes, because the component proteins are identical, all sequences in the monomer MSAs will be linked with themselves, which results in the complete use of both genomics and metagenomics databases, and substantially larger structure

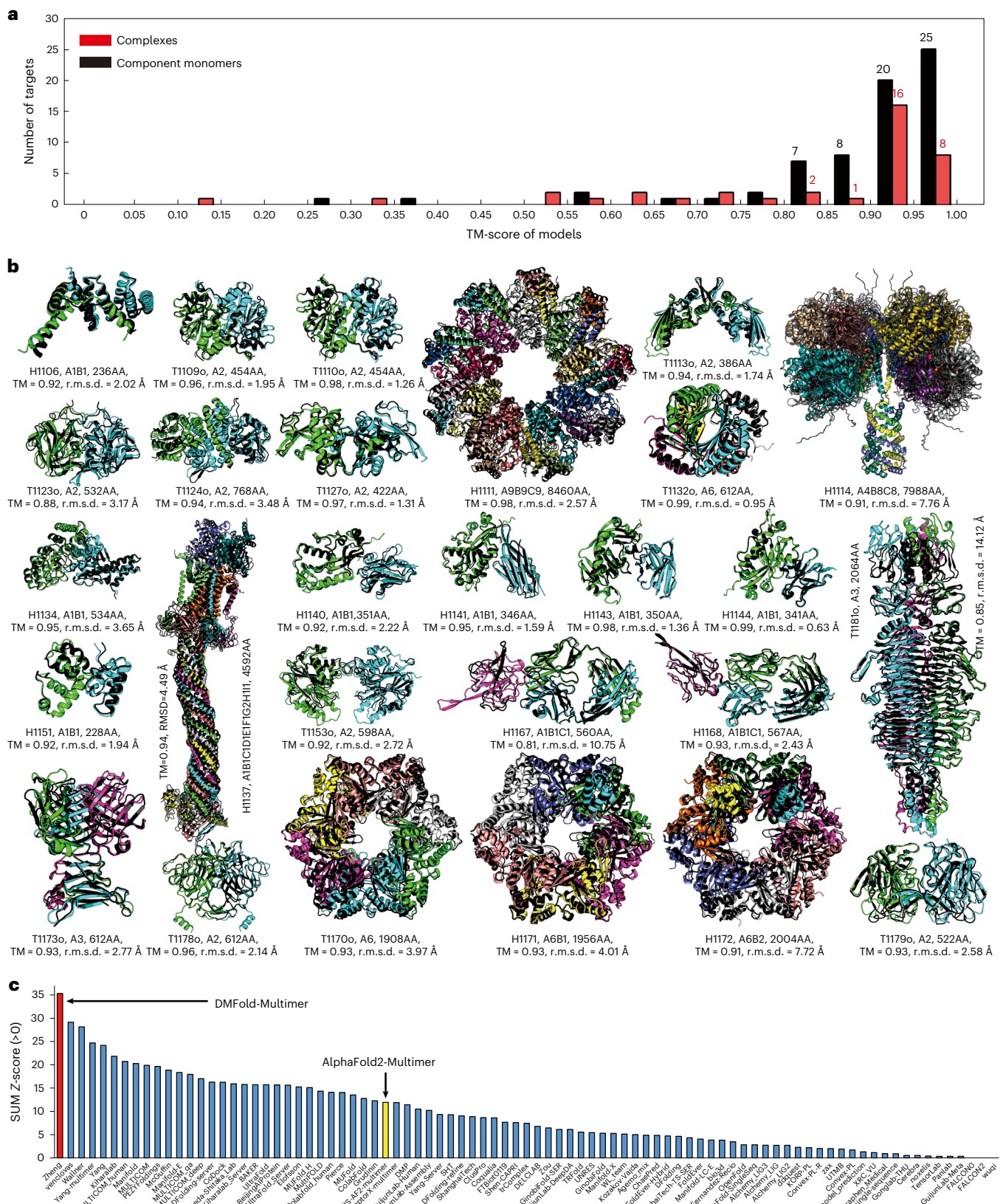

**Fig. 6 | Performance of the DMFold/DMFold-Multimer pipeline for protein complex structure prediction in the CASP15 experiment. a**, Histogram of the TM-scores of structural models by DMFold-Multimer on the 38 complex targets that have their experimental structure released. **b**, The first models produced by DMFold-Multimer superposed on the experimental structures for the 27 complex targets with TM-scores (TM) >0.8, where the component monomers of the predicted models are shown in distinct colors with the experimental structures marked in black. **c**, Sum of Z-scores on 41 multimeric targets for the 87 registered CASP15 assembly groups, with data taken from the CASP15 webpage. DMFold-Multimer (registered as Zheng) and the public March-2022 version of the AlphaFold2-Multimer server (registered as NBIS-AF2-multimer) are marked in red and yellow, respectively. r.m.s.d., root mean square deviation.

improvement of DMFold-Multimer over AlphaFold2-Multimer. We expect that the performance of DMFold-Multimer on heteromeric complexes can be further improved in future using more comprehensive taxonomic annotation databases, or through the development of new sequence linking algorithms covering metagenomic sequences.

The positive impact of multimeric MSAs provided by DeepMSA2 manifests itself mainly in the interchain orientation/distance map predictions. As a case study, in Fig. 5c we present an example from T0988o, which is a homo-trimer containing three protein chains with 612 residues in total. AlphaFold2-Multimer creates an incorrect model with a low TM-score of 0.18. A likely cause is that the AlphaFold2-Multimer default MSA pipeline only detects eight homologous sequences for each component protein, leading to a poor prediction of the intrachain distance map (MAE = 5.93 Å) used for protein monomer structure inference (Fig. 5d), thus resulting in a poor monomer structure model with a monomer TM-score of 0.20. By contrast, DeepMSA2 detects 71 homologous sequences, which results in a more accurate distance map (MAE = 0.81 Å, Fig. 5e) and a more accurate structure model (monomer TM-score = 0.95) for each chain. At the complex level, DMFold-Multimer generates a high-quality complex structure model for T0988o with a TM-score of 0.96. Figure 5f shows another illustrative case from T1038o, which is a homo-dimeric complex with 796 residues in total. Although AlphaFold2-Multimer generates a good-quality monomer model with a TM-score of 0.91, the quaternary orientation of the complex is completely wrong resulting in a poor complex TM-score of 0.53. This is mainly due to the poor MSA constructed by the AlphaFold2-Multimer pipeline (with Neff = 0.2), which results in a very low interchain distance prediction accuracy (MAE = 4.11 Å; Fig. 5g). For this example, DeepMSA2-Multimer creates a deeper MSA with Neff = 1.4, which results in a high-accuracy prediction for both intrachain (MAE = 1.11 Å) and interchain (MAE = 1.80 Å) distance maps (Fig. 5h). As a result, DMFold-Multimer creates a much more accurate complex structure model with a complex TM-score of 0.91. These results show that correct construction of multimeric MSAs is critical to both quaternary restraints and final model predictions.

## Blind test of DeepMSA2/DMFold-Multimer in CASP15 experiment

As a blind test, the DeepMSA2/DMFold-Multimer pipeline participated in the community-wide CASP15 experiment held in 2022 for complex structure prediction. This experiment contained 47 complex targets, each with 2–27 component chains. In Fig. 6a, we show a histogram of TM-scores for the structural models by DMFold-Multimer for the 38 complexes that have experimental structure released, where DMFold-Multimer created models with an average TM-score of 0.83, and 36 of the 38 complexes have a TM-score above 0.5. Despite the promising result, the TM-score of the complex models is still considerably lower than that for the corresponding monomer models (TM-score = 0.89), suggesting that interchain orientation is still a challenging issue in quaternary protein structure prediction.

In Fig. 6b, we present the superposition of the DMFold-Multimer models on the experimentally solved structures for 27 complex targets for which the predictions have a TM-score >0.8. These include seven large-size complexes from H1111, H1114, H1137, T1170o, H1171, H1172 and T1181o, the sequences of which contain 8,460, 7,988, 4,592, 1,908, 1,956, 2,004 and 2,064 residues, where DMFold-Multimer constructed impressive complex models with TM-scores of 0.98, 0.91, 0.94, 0.93, 0.93, 0.91 and 0.85, respectively. Notably, the three largest targets are all heteromeric complexes with stoichiometry variable of 'A9B9C9', 'A4B8C8' and 'A1B1C1D1E1F1G2H1I1', respectively, where DMFold-Multimer constructed high-accuracy models with TM-score >0.9 for all of them. These results demonstrate the ability of DMFold-Multimer to model large protein complexes, which has been a long-term challenge for traditional quaternary structure modeling approaches[23].

In Fig. 6c and Supplementary Table 9, we also list a comparison of DMFold-Multimer (named Zheng) with 86 other methods participated in the CASP15 Multimeric Modeling Section. DMFold-Multimer outperformed all other groups in terms of the sum of Z-score, which was calculated by the CASP assessors based on a combination of TM-score, LDDT, ICS and Interface Patch Score; where TM-score and LDDT measure the global fold quality and ICS and Interface Patch Score assess the protein interface modeling quality of protein complexes. Overall, DMFold-Multimer achieved a cumulative Z-score of 35.30, which is nearly three times higher than that of the 'NBIS-AF2-multimer' group (that is, the public March-2022 v.2.2.0 of the AlphaFold2-Multimer server run by the Elofsson Lab on CASP15 targets, which achieved a cumulative Z-score of 12.27) and 21.1% higher than the second-best performing group (29.15). A breakdown of the component score comparisons between DMFold-Multimer and AlphaFold2-Multimer is shown in Supplementary Table 10.

In Supplementary Fig. 7, we show three illustrative examples from targets H1140, H1141 and H1144, which are all nanobody–antigen complexes. Nanobodies are single-domain antibodies that initiate critical immune reactions by interacting with antigens[24], where these targets represent three typical interaction modes of nanobodies with the same mouse 2′,3′-Cyclic-nucleotide 3′-phosphodiesterase. As shown in Supplementary Fig. 7, the complex models by AlphaFold2-Multimer (the NBIS-AF2-multimer group) have a relatively low TM-scores (<0.7), whereas DMFold-Multimer created excellent predictions for the three complexes with TM-scores of 0.92, 0.95 and 0.99, respectively. Accordingly, the ICS F1 scores of the DMFold-Multimer models (0.51, 0.79 and 0.74) are much higher than those for AlphaFold2-Multimer (0.02, 0.06 and 0.09), suggesting that correct construction of the DeepMSA2-Multimer MSAs has largely enhanced the modeling of quaternary chain interactions in these immune protein–antigen complex targets. In Supplementary Fig. 8, we investigate the target H1144 to further examine the difference in the two programs. In this example, DMFold-Multimer utilizes a multi-MSA pairing strategy to create 25 paired MSAs, with the best model of TM-score (0.99) coming from the MSA with the highest Neff (16.3). By contrast, AlphaFold2-Multimer uses a single MSA with Neff = 8.1 which resulted in no models with a TM-score >0.8. Nevertheless, DMFold-Multimer could not fold all the nanobody–antigen complexes in CASP15. Supplementary Fig. 9 presents a failed nanobody–antigen case from target H1142 in which no correct model was created by DMFold-Multimer despite the use of multiple MSAs. A detailed discussion on the failure and success of DMFold-Multimer on the nanobody–antigen complex structure modeling is summarized in Supplementary Discussion Text 2.

## Discussion

With the rapid progress of deep machine-learning techniques, MSAs have become increasingly essential to modern protein structure predictions. Built on iterative alignment searches through multiple genome and metagenome sequence databases, we have developed a hierarchical pipeline, DeepMSA2, for protein monomer and multimer MSA construction. Large-scale tests show that DeepMSA2 can be used to substantially improve the accuracy of protein structure predictions.

Compared with existing MSA construction methods, one of the major advantages of DeepMSA2 lies in the iterative search and model-based preselection strategy, which can result in MSAs with more balanced alignment coverage and homologous diversity. The iterative searching strategy also allows for the exploration of multiple in-house metagenome sequence databases, which helps increase the diversity and coverage of the resulting MSA. Detailed benchmark data show that the evolutionary/coevolutionary information derived from such MSAs can clearly improve the accuracy of structure template recognition and deep learning distance/orientation restraint predictions. By integrating DeepMSA2 with the state-of-the-art AlphaFold2 modeling approach, DMFold can improve the TM-score of AlphaFold2

models by 5% for FM domains that lack homologous templates in the PDB. Our application of DMFold on the human proteome has resulted in an 11% increase in pLDDT score for the 5,042 difficult proteins for which AlphaFold2 failed to create confident models, thus substantially expanding the range of human proteins for which actionable structure predictions can be provided.

Utilization of DeepMSA2-Multimer MSAs has also resulted in substantial improvements in multichain protein complex structure prediction, where a 12.2% increase in TM-score is obtained by DMFold-Multimer over the default AlphaFold2-Multimer for the 54 complexes from the CASP13 and CASP14 experiments. In the most recent community-wide blind test of CASP15, DMFold-Multimer achieved the highest modeling accuracy for complex structure prediction, with an average TM-score 15.4% higher and average ICS score 27.5% higher than the public March 2022 v.2.2.0 of the AlphaFold2-Multimer server run by the Elofsson Lab (registered as NBIS-AF2-multimer), according to the assessor's criteria. Notably, DMFold-Multimer constructed high-quality models for massive oligomer complexes up to 8,460 amino acids with a TM-score of 0.98, highlighting its ability to model large protein complex structures, addressing a persistent problem that has challenged traditional protein quaternary structure prediction[23].

Despite the impressive improvements in performance, some challenges remain for DeepMSA2. One key area with likely room for improvement is in the modeling of heteromeric complexes, which show smaller improvements (relative to AlphaFold2) than homomer complexes in our internal testing. Because current multimer MSAs are built from monomer MSAs, a fundamental challenge to be addressed is how to effectively link the sequences of different component MSAs to form optimal multimeric MSAs of interacting homologous sequences. The current sequence linking mechanism, which is based on species annotation, works only on genomic sequences, and thus the highly informative homologous sequences from metagenomics databases cannot be fully utilized to guide multichain structural assembly. For homomer complexes, the current approach simply links all sequences of monomer MSAs to themselves. However, not all homologous sequences interact with themselves and an approach correctly linking the interacting homologs, but not noninteracting homologs (for example, based on protein–protein interaction predictions), may help further improve homomer MSA construction. The identification of robust methods for optimal construction of paired MSAs will likely be of great value to future efforts to optimize predictions regarding heterologous contacts between proteins, as well as for related tasks such as classifying arbitrary protein pairs as interacting versus noninteracting. In addition to MSA pairing, another potential area for growth is to retrain AlphaFold2-like models making explicit use of the more informative monomer MSAs; this may help address the sequence pairing issues directly through an integrated network learning process.

In addition, stoichiometry information (the number of copies of each component chain) for the complex is required before implementing the DMFold-Multimer pipeline, which may limit the usefulness of the method in practical applications. Including a deep learning-based stoichiometry predictor[25,26] based on the query sequences and evolutionary signals to DMFold-Multimer pipeline may be part of the solution to alleviating this limitation. Furthermore, whether the current DeepMSA2/DMFold approach could be extended to RNA and RNA–protein complex structure prediction is also a topic to explore in our ongoing research, where both limitations on the sparse availability of RNA sequence and structure databases compared with proteins need to be overcome.

More generally, the strong performance that we observe through the DeepMSA2/DMFold pipeline demonstrates that the protein structure prediction problem is not 'solved'. Substantial room for improvement over the current state-of-the-art still exists, particularly for proteins with few identifiable sequence homologs, and those involved in multiprotein complexes. The DeepMSA2/DMFold approach provides substantial advances in the prediction of some such difficult targets, showing additional evidence that optimization of the information content of input to deep learning-based protein structure prediction methods must be considered with as much care as the design of the predictor itself.

## Online content

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

## Methods

### Benchmark dataset preparation

**Monomer proteins from CASP.** Some 293 domains from monomer targets in the CASP experiments were collected to benchmark the effect of DeepMSA2 on monomer protein structure prediction. The CASP experiments often classify the domains as template-based modeling (TBM)-easy, TBM-hard, TBM/FM and FM. To simplify the data analyses, we merged TBM-easy and TBM-hard domains as TBM domains, and TBM/FM and FM domains as FM domains in this study. In our benchmarks, 48 FM domains and 64 TBM domains came from CASP13; 37 FM domains and 50 TBM domains were taken from CASP14; and 47 FM domains and 47 TBM domains were from CASP15 (Supplementary Table 11).

**Multimer protein complexes from CASP.** Fifty-four protein complex targets were collected from CASP13 and CASP14, including 14 heteromer and 40 homomer complexes. Here we use the stoichiometry variable to represent the type of a complex; that is, with the alphabet representing different protein chains and the number after each letter indicating the number of copies of the corresponding component protein chain in the complex. For example, 'A3B2C2' means the complex contains three different protein chains, A, B and C, and there are three copies of protein A, two copies of protein B and two copies of proteins C in this complex. For homomers, the dataset contains twenty-two 'A2' (dimer), thirteen 'A3' (trimer), four 'A4' (tetramer) and one 'A8' (octamer) complexes. For heteromers, nine 'A1B1', three 'A2B2', one 'A3B1' and one 'A1B1C1D1E1' complexes are in the dataset (Supplementary Table 7).

**Human proteome.** The human proteome dataset contains more than 20,000 proteins or peptides with lengths between 2 and 34,350 amino acids collected from UniProt (https://www.uniprot.org/uniprotkb/?facets=reviewed%3Atrue&query=proteome%3AUP000005640). In 2021, DeepMind released the AlphaFold2 model database, AlphaFold2 DB[22], which contains structure models predicted by AlphaFold2 for several reference proteomes, including the human proteome. However, only around 70% (13,838) of human proteins in AlphaFold2 DB have confident predictions with pLDDT ≥ 0.7. From the remaining 6,757 proteins for which AlphaFold2 failed to create confident folds, we selected the 5,042 proteins with lengths <800 amino acids for remodeling by DMFold.

### Number of effective sequences in the MSA

To quantify the sequence diversity of an MSA, we define the number of effective sequences (Neff) as follows:

$$\text{Neff} = \frac{1}{\sqrt{L}} \sum_{n=1}^{N} \frac{1}{1 + \sum_{m=1,m\neq n}^{N} I\left[S_{m,n} \geq 0.8\right]} \qquad (1)$$

where $L$ is the length of the query sequence, $N$ is the number of sequences contained in the MSA, $S_{m,n}$ is the sequence identity between the $m$th and $n$th sequences, and $I[]$ represents the Iverson bracket, which takes the value $I[S_{m,n} \geq 0.8] = 1$ if $S_{m,n} \geq 0.8$, and 0 otherwise.

Based on the definition in equation (1), MSAs with a more diverse set of sequence pairs with sequence identity <0.8 have the term $\sum_{m=1,m\neq n}^{N} I[S_{m,n} \geq 0.8]$ closer to 0, and thus result in a higher Neff value given the same number of sequences ($N$). In case that all $N$ sequences in an MSA are diverse (pairwise sequence identity <0.8), the term $\sum_{m=1,m\neq n}^{N} I[S_{m,n} \geq 0.8] = 0$, and then Neff will be $N/\sqrt{L}$. In other words, given a Neff cutoff $\text{Neff}_{\text{cut}}$, the minimal number of diverse sequences needed for modeling can be roughly estimated by

$$N_{\min} = \text{Neff}_{\text{cut}} \times \sqrt{L} \qquad (2)$$

It is generally believed that MSAs with more diverse sequences and higher alignment coverages can provide more evolutionary and coevolutionary information and thus better assist deep learning protein structure prediction. To quantitatively evaluate that belief based on our data, in Supplementary Fig. 10a we plot the TM-score of DMFold models versus Neff values for the 62 monomer FM domains in CASP13–CASP15. Although higher Neff values tend to correspond to models with higher TM-scores, there is no clear quantitative threshold of Neff corresponding to the absolute success of structure modeling. Following the general trend, we can provide two approximate thresholds, $\text{Neff} = 2^0$ and $\text{Neff} = 2^4$, which are roughly associated with three TM-score territories; that is, the average TM-scores with Neff in $[0, 2^0]$, $(2^0, 2^4]$ and $(2^4, \infty)$ are roughly <0.70, ~0.85 and >0.90, respectively. Thus, following equation (2), approximately at least 10 (= $2^0 \times \sqrt{100}$) or 160 (= $2^4 \times \sqrt{100}$) diverse sequences are required for good- or high-accuracy modeling of a 100-residue protein, respectively.

In Supplementary Fig. 10b, we also present a comparison of TM-score versus alignment coverage, which is defined as the average rate of aligned residues on the query sequence across all homologous sequences in the MSA, for the same set of recent CASP targets. The data show no obvious correlation between TM-score and coverage of the MSA. It is obvious that an MSA with coverage that is too short (for example, with the alignment focused only on the N-terminal tail) is useless for deducing the coevolutionary signal of the entire protein sequence. Our data suggest, however, that the final performance of structure prediction does not depend on the alignment coverage as long as the alignment covers a reasonable region of the query sequence (for example, more than ~60% in our case).

### Genome database collection

Three genomic sequence databases, Uniclust30[27], UniRef30[27] and Uniref90[28], which are all based on UniProtKB[29], are utilized in the DeepMSA2 pipeline (details are given in Supplementary Table 12).

**Uniclust30/UniRef30.** The Uniclust30/UniRef30 database is an HHblits-style[8] hidden Markov model (HMM) database that clusters UniProtKB[29] sequences at the level of 30% pairwise sequence identity by MMseqs2[9]. For each cluster, an 'A3M' formatted MSA generated by Clustal Omega[30] and the corresponding HMM are provided for HHblits. Whereas Uniclust30 is the version of the database generated before 2019, UniRef30 is the one generated after 2019. In total, Uniclust30 contains 124 million sequences in 15 million clusters, and UniRef30 provides 231 million sequences in 25 million clusters.

**Uniref90.** Uniref90 provides sequences from the UniProtKB clustered at 90% pairwise sequence identity by MMseqs2. Unlike Uniclust30 and UniRef30 (which are HMM databases), Uniref90 is a flat sequence database. For each cluster, only the representative sequence of the cluster is kept in the database. Thus, there are 109 million protein sequences/clusters in the Uniref90 database.

### Metagenomics genome database collection

Six metagenomics sequence databases are utilized in DeepMSA2. These include three third-party databases (Metaclust[31], BFD[31] and Mgnify[32]) and three in-house databases (TaraDB[21], MetaSourceDB[6] and JGIclust). The three in-house databases, which were built using data collected from the European Bioinformatics Institute (EBI) Metagenomics project[33] and the Joint Genome Institute (JGI)[34], contain in total 35.6 billion nonredundant sequences, which are approximately 11 times as large as the three third-party metagenomics databases used (~3.2 billion).

**Metaclust.** Metaclust was created by clustering and assembling 1.59 billion protein sequence fragments predicted by Prodigal[35] in ~2,200 metagenomics and meta-transcriptomic datasets that came from the JGI[34]. The 1.59 billion metagenomics sequences were clustered

with 50% sequence identity at 90% coverage, yielding 712 million clusters and the corresponding nonredundant sequences.

**BFD.** The BFD is an HHblits-style HMM database that was created by clustering 2.5 billion protein sequences from UniProtKB[29], Metaclust[31], soil reference catalog and marine eukaryotic reference catalog assembled by Plass[36]. BFD was clustered by MMseqs2[9] with 30% pairwise sequence identity, and only the clusters that have more than three sequence members were kept in the database. In total, 66 million clusters and 2.2 billion genomics/metagenomics sequences are collected in the BFD database.

**Mgnify.** The Mgnify metagenomics database was collected by the EBI Metagenomics project[32,33] and was clustered by MMseqs2 using coverage and sequence identity threshold set at 90%. Similar to Uniref90, for each cluster, only the representative sequence was kept in Mgnify database, leading to 305 million metagenomics sequences.

**TaraDB.** We obtained 245 metagenomics sequencing runs from the 'Tara Oceans' project hosted on EBI Metagenomics (https://www.ebi.ac.uk/metagenomics/studies/ERP001736). To obtain protein sequences for Tara metagenome database, a pipeline combining raw reads assembly, open reading frames (ORFs) identification and redundant sequence trimming approaches were implemented[21]. The raw read sequences were assembled by MEGAHIT v.1.0 to contigs and only contigs with >500 nucleotides are selected. Next, Prodigal (v.2.6) was used with parameters '-c –m p meta' to identify ORFs from metagenome data and translate the gene to protein productions. Finally, CD-HIT (v.4.6)[37] was utilized to cluster protein sequences in each sample, and the sequence identity threshold was set to 95% to remove the identical sequence. In total, the Tara metagenome database contains 121 million protein sequences.

**MetaSourceDB.** MetaSourceDB was used in our previous MetaSource research[6], which collects metagenome data from four large environmental biomes of the EBI database (https://www.ebi.ac.uk/metagenomics/). Those four biomes, including 'fermentor', 'soil', 'lake' and 'gut', cover all typical biomes of the EBI database. In total, 1,705 high-quality samples were selected, assembled and clustered by the similar pipeline used in TaraDB. In addition to Prodigal, FragGeneScan[38] (v.1.20) was also used to predict ORFs from assembled contigs to avoid missing the short sequences. Overall, 805, 4,170, 1,290 and 12,811 million protein sequences are collected for the 'fermentor', 'soil', 'lake' and 'gut' biomes, respectively, resulting in 19.1 billion proteins contained in MetaSourceDB.

**JGIclust.** We collected ~25,000 metagenomics and meta-transcriptomic samples from the JGI[34]. For each project, the assembled protein sequences ('*.assembled.faa') were downloaded and clustered with 90% sequence identity at 90% coverage by MMseqs2. For each cluster of one project, only the representative sequence was kept in the in-house JGIclust database. A total of 16.4 billion metagenomics and meta-transcriptomic sequences are contained in the JGIclust database.

## DeepMSA2-Monomer pipeline for monomeric MSA construction

Given the complexity and huge size of the protein sequence databases, there is no single MSA construction tool that can quickly search through all sequence databases and create reliable MSAs for different targets. To address this issue, DeepMSA2 utilizes a multi-MSA generation step to create a diverse set of MSAs using the dMSA, qMSA and mMSA programs, followed by a MSA ranking step based on rapid deep learning structure prediction (Fig. 1a).

**dMSA.** The dMSA algorithm used in DeepMSA2 is modified from our previous MSA generation tool, DeepMSA. Here, dMSA generates up to

three MSAs by a three-stage procedure that uses HHblits[8], Jackhmmer[39] and HMMsearch[39] to iteratively search the genomic and metagenomics sequence databases. In stage 1, HHblits is used to search Uniclust30 with the parameters '-diff inf -id 99 -cov 50 -n 3'. In stage 2, Jackhmmer is used to search against Uniref90 with parameters '-N 3 -E 10 --incE 1e-3' to pick up potentially homologous sequences. Instead of directly using the alignment generated by Jackhmmer, the full-length sequences according to the Jackhmmer raw hits are collected from Uniref90. These full-length sequences are clustered by kClust into sequence clusters by 30% sequence identity cutoff. Next, Clustal Omega[30] is used to realign sequences within each cluster into aligned sequence profiles. Those sequence profiles are then converted to a custom HHblits-style database using the 'hhblitdb.pl' script from the HH-suite package. HHblits is again applied to search this custom database using the same search parameter as in stage 1 but starting from the stage 1 sequence MSA as input. In stage 3, the MSA from stage 2 is converted to a HMM by HMMbuild from the HMMER package. This HMM is searched against the Metaclust metagenome database by HMMsearch, using parameters '-E 10 --incE 1e-3'. Similar to stage 2, sequence hits from HMMsearch are built into a custom HHblits database. The MSA from the previous stage is used to search against this new custom HHblits database to derive the stage 3 MSA. In addition, to speed up the custom database construction and filter out the noisy raw sequences picked up by Jackmmer and HHMsearch in stages 2 and 3, respectively, a BLAST filter is applied to the raw sequences obtained from Uniref90 before kClust clustering. Here, PSIBLAST[18] is used to rank the homologous relation between the raw hits and the query sequence by e-value, and up to 30,000 top-ranked raw hits will be used in the kClust clustering step. Based on our benchmark, by adding this BLAST filter, the AlphaFold2 modeling accuracy can be slightly increased compared with that without the filter in the CASP14 monomer dataset (Supplementary Table 13). The dMSA construction will be stopped at any searching stage whenever the Neff value is >128. Thus, at most three MSAs will be generated by dMSA.

**qMSA.** qMSA is composed of four stages that perform HHblits (v.2), Jackhmmer, HHblits (v.3) and HMMsearch searches against Uniref30, Uniref90, BFD and Mgnify databases, respectively. In the BFD database-searching step, HHblits (v.3) is utilized with parameters '-diff inf -id 99 -cov 40 -n 3 -e1'. Similar to dMSA stages 2 and 3, the sequence hits from Jackhmmer, HHblits (v.3) and HMMsearch in stages 2, 3 and 4 of qMSA are converted to HHblits (v.2) formatted databases, against which the HHblits (v.2) search based on the MSA input from the previous stage is performed. As with dMSA, the searching will stop when the MSA from the current stage of qMSA has Neff > 128, resulting in up to four MSAs created by the qMSA method.

**mMSA.** In mMSA, the qMSA stage 3 alignment is used as a probe by HMMsearch using parameters '-E 10 --incE 1e-3' to search through a metagenomics database combining JGIclust, TaraDB and MetaSourceDB, with the resulting sequence hits converted to a raw sequence database. This mMSA database is then used as the target database, which is searched by HHblits (v.2) with three seed MSAs (MSAs from dMSA stage 2 and qMSA stages 2 and 3) to derive three new MSAs. The mMSA program will not be used if both dMSA and qMSA stopped at stage 1, which means that the number of detected homologous sequences is sufficiently reliable from genomic sequence databases.

**Final MSA selection.** A simplified version of AlphaFold2 is applied here to rank the MSAs generated by dMSA, qMSA and mMSA, where the template detection module is turned off and the embedding parameter is set to one in AlphaFold2 for rapid model generation. Up to ten MSAs are collected from the MSA generation step, where each of the MSAs is fed into the modified AlphaFold2 program to create five structure models. For a given MSA, the highest pLDDT score among the five predicted models is assigned as the rank score of the MSA. The MSA with

the highest rank score among all created MSAs is returned as the final selected MSA, reflecting an optimization of the information content contributing to protein structure prediction.

### DeepMSA2-Multimer pipeline for multimeric MSA construction

DeepMSA2 for multimeric MSA construction contains four steps: (1) monomeric MSA generation, (2) monomeric MSA pairing, (3) joint MSA creation by sequence linking, and (4) multimeric MSA ranking and selection (Fig. 1b).

**Monomeric MSA generation.** The abovementioned DeepMSA2-Monomer pipeline is used to create monomeric MSAs for each of the component chains. However, instead of returning only one top-ranking MSA, up to ten MSAs are kept for each chain, to facilitate the modeling of quaternary orientations of between different component chains.

**MSA pairing.** Two types of complexes are considered in DeepMSA2. For homomeric complexes in which all component chains are identical, all of the monomer MSAs are utilized and the multimeric MSAs are created by concatenating each of the monomer MSAs $n$ times side-by-side, where $n$ is the number of monomer chains (Supplementary Fig. 11). For heteromeric complexes, the top $M$ MSAs are selected for each monomer chain so that $M^N$ distinct paired MSAs can be created for the complex, where $N$ is the number of distinct chains in the complex. To avoid an impractically long MSA construction time, $M$ is set as the maximal value to satisfy $M^N \leq 100$. For example, for a complex containing three different protein chains (A2B2C1, $N = 3$), $M$ will be set to 4 ($4^3 \leq 100$) (Supplementary Fig. 12a). In other words, for each component chain in this complex, we select four top-ranked monomer MSAs and build paired MSAs for the complex with 64 different combinations of those monomer MSAs. Normally, $M^N$ ranges from 50 to 100 for different kinds of heteromer complexes.

**Sequence linking.** For a given set of $M^N$ paired monomeric MSAs, for example (MSA $-1_{i_1}$, MSA $-2_{i_2}$, ..., MSA $-N_{i_N}$) with $1 \leq i_1, i_2, ..., i_N \leq M$, the sequences from the monomeric MSAs are concatenated into a multimeric MSA as follows (Supplementary Fig. 12b). First, the sequences in each monomeric MSA are grouped based on the UniProt annotated species. The sequences in each group are then ordered based on the sequence identity to the query sequence. To properly capture orthologous signals of interchain coevolution, the top sequences of different monomeric MSAs belonging to the same species group are linked together side-by-side to form a composite sequence in the multimeric MSA. In cases where one of the monomeric MSAs is missing for a specific species, which appear in more than one other chains, the component chain is padded with gaps in the composite sequence with other linked chains having that species. Finally, the unlinked sequences in the monomeric MSAs are padded below the linked sequences. This composite linking step is applied only to heteromeric complexes, as the MSAs for homomeric complexes are constructed by simply concatenating the same monomer MSA multiple times as shown in Supplementary Fig. 11.

**MSA selection.** Of the $M^N$ concatenated MSAs formed from the MSA paring procedure, 25 top MSAs are returned from the DeepMSA2-Multimer pipeline based on the $M$-score:

$$M\text{-score} = \text{Neff}\left(\frac{\sum_{i=1}^{N} n_i \times \text{pLDDT}_i}{\sum_{i=1}^{N} n_i}\right) \quad (3)$$

where Neff is the depth of the concatenated MSA calculated based on equation (1). $n_i$ is the copy number of $i$th component chain in the

complex where pLDDT$_i$ is the pLDDT score of chain-$i$ taken from the monomeric MSA generations. Again, this step is designed for heteromeric complexes, whereas for homomeric complexes, all ten MSAs from self-concatenation are returned.

### Protein tertiary and quaternary structure prediction

**AlphaFold2 and AlphaFold2-Multimer programs.** In the standard AlphaFold2 program[20], an end-to-end network architecture is implemented on predicting 3D structure of monomeric proteins from an MSA and homologous templates. AlphaFold2-Multimer[4] was extended from AlphaFold2 protocol for quaternary structure prediction by training the networks on protein complex structures.

**DMFold.** DMFold-Monomer (or DMFold) is designed for modeling structure of monomer proteins by combing the DeepMSA2 and AlphaFold2 pipelines. The major difference between DMFold and AlphaFold2 is that the MSAs in AlphaFold2 are regenerated by DeepMSA2.

**DMFold-Multimer.** The DMFold-Multimer pipeline utilizes AlphaFold2-Multimer to generate complex structure models, but with $k$ multimer MSAs from DeepMSA2-Multimer as the input matrix, where $k = 25$ for heteromer and $k = 10$ for homomer complexes. For each multimer MSA, 25 models are generated. Finally, the resulting 625 (or 250 for homomer) complex models are ranked by the predicted TM-scores[4,40], and the top five complex models are selected as the final set of models.

We note that we did not retrain the AlphaFold2 or AlphaFold2-Multimer network models with DeepMSA2 MSAs in the DMFold or DMFold-Multimer pipeline, because one focus of this study is on comparing the impact of the MSAs on protein structure prediction and making a fair comparison with AlphaFold2 and AlphaFold2-Multimer. Meanwhile, because the original AlphaFold2 training sets contain proteins with various MSA qualities—for example, with Neff ranging from low to high values—it is expected that the retraining of AlphaFold2 models on a set of improved MSAs should have a minimal impact (if any) on the final model quality. Thus, for the calculations shown here, we simply used the DeepMind pretrained AlphaFold2 and AlphaFold2-Multimer models and parameters when implementing DMFold and DMFold-Multimer programs.

### DeepPotential for residue–residue restraint prediction

DeepPotential[19] is a deep learning algorithm predicting distance and geometry restraints of proteins based on MSAs (Supplementary Fig. 13). Given an MSA, two major pair features are extracted, including raw coupling parameters from the pseudo likelihood maximized (PLM) 22-state (the 20 standard amino acids, the nonstandard amino acid type and the gap state) Potts model and the raw mutual information (MI) matrix. The PLM feature minimizes a loss function defined by

$$\mathcal{L}_{\text{PLM}} = -\sum_{i=1}^{L} \sum_{n=1}^{N} \ln \frac{\exp\left(h_i(\sigma_n^i) + \sum_{j=1, j\neq i}^{L} P_{i,j}(\sigma_n^i, \sigma_n^j)\right)}{\sum_{q=1}^{Q} \exp\left(h_i(q) + \sum_{j=1, j\neq i}^{L} P_{i,j}(q, \sigma_n^j)\right)} + \lambda_{\text{single}} \sum_{i=1}^{L} \|h_i(\sigma_n^i)\|_2^2$$
$$+ \lambda_{\text{pair}} \sum_{i=1}^{L} \sum_{j=1}^{L} \|P_{i,j}(\sigma_n^i, \sigma_n^j)\|_2^2 \quad (4)$$

where $L$ is the length of the protein and $N$ is the number of aligned sequences in the MSA, $\sigma_n^i$ indicates the amino acid type of $n$th sequence and $i$th position in the MSA; and $h$ and $P$ are field and coupling parameters respectively. $Q = 22$, representing 20 types of regular amino acids, plus the unknown residue type state and the gap state. Additional L2 regularization terms are also added to avoid possible overfitting, where $\lambda_{\text{single}} = 1$ and $\lambda_{\text{pair}} = 0.2 \times (L - 1)$ are the regularization coefficients. The MI feature of residue pair $i$ and $j$ is defined by:

$$M_{i,j}(k,l) = f_{i,j}(k,l) \ln \frac{f_{i,j}(k,l)}{f_i(k)f_j(l)} \qquad (5)$$

where $f_i(k)$ is the frequency of a residue type $k$ at position $i$ of the MSA, $f_{i,j}(k,l)$ is the co-occurrence of two residue types $k$ and $l$ at positions $i$ and $j$. Complementary information—that is, conditional and marginal relationships between residues—can be extracted from the MSAs by PLM and MI features, respectively.

In addition, sequential features, such as self-mutual information feature, field parameter of the Potts model, one-hot sequence feature and HMM profiles, are also considered as the inputs of DeepPotential. The sequential features and pair features are fed into deep convolutional neural networks separately, where each of them is passed through a set of 10 one-dimensional and 10 two-dimensional residual blocks, which are then tiled together. The feature representations are used as the inputs of another fully residual neural network containing 40 two-dimensional residual blocks which output interresidue distance terms (Supplementary Fig. 13).

To assess the accuracy of the distances predicted by DeepPotential relative to experimental results, the MAE of the top $5L$ ($L$ is the protein length, in amino acids) long-range ($|i - j| \geq 24$) predicted distances is considered:

$$MAE = \frac{1}{5L} \sum_{(i,j)}^{5L} \left| d_{i,j}^{pred} - d_{i,j}^{exp} \right| \qquad (6)$$

where $d_{i,j}^{exp}$ is the Cβ–Cβ distance between residue $i$ and $j$ in the experimental structure, and $d_{i,j}^{pred}$ is the predicted Cβ–Cβ distance between residue $i$ and $j$ from DeepPotential; the latter is estimated as the middle value of the bin with the highest probability.

### Model quality assessment

TM-score[40] is used in the work to assess the model quality for both monomer and complex structures of proteins. For calculating TM-score, US-align[41] is utilized here, with the commands 'US-align monomer-model.pdb native.pdb' for protein monomer and 'US-align complex-model.pdb native.pdb –ter 0 –TMscore 6' for protein complexes. All data statistical analyses are done by R (v.4.1.2).

### Reporting summary

Further information on research design is available in the Nature Portfolio Reporting Summary linked to this article.

### Data availability

The third-party databases used in this work, Uniclust30 (UniRef30), BFD, Uniref90, Metaclust and MGnify, are available at https://gwdu111.gwdg.de/~compbiol/uniclust/, https://bfd.mmseqs.com/, https://ftp.uniprot.org/pub/databases/uniprot/current_release/uniref/uniref90/, https://metaclust.mmseqs.org/ and http://ftp.ebi.ac.uk/pub/databases/metagenomics/peptide_database/, respectively. All CASP benchmark data used in this work are available at https://zhanggroup.org/DMFold/ (or https://zenodo.org/record/8371924). The structure modeling results on 5,042 human proteome proteins are freely available at https://zhanggroup.org/DMFold/human (or https://zenodo.org/records/10099696) for academic use. Source data are provided with this paper.

### Code availability

The online servers of DeepMSA2/DeepMSA2-Multimer and DMFold/DMFold-Multimer are freely available at https://zhanggroup.org/DeepMSA/ and https://zhanggroup.org/DMFold, respectively. The standalone packages of DeepMSA2/DeepMSA2-Multimer and DMFold/DMFold-Multimer are freely available at https://zhanggroup.org/DeepMSA/download (or https://zenodo.org/record/10092418) and https://zhanggroup.org/DMFold/download (or https://zenodo.org/records/10092882), respectively, for academic use.

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

### Acknowledgements

DeepMSA2/DeepMSA2-Multimer and DMFold/DMFold-Multimer were trained using the Extreme Science and Engineering Discovery Environment (XSEDE), which is supported by the National Science Foundation (grant no. ACI1548562). We are grateful to J. Poisson, B. Palen and the staff of the University of Michigan Advanced Research Computing team for technical assistance. This work is supported in part by the National Institute of General Medical Sciences (grant nos GM136422 and S10OD026825 to Y.Z.), the National Institute of Allergy and Infectious Diseases (grant no. AI134678 to L.F. and Y.Z.) and the National Science Foundation (grant nos IIS1901191 and DBI2030790 to Y.Z., and grant no. MTM2025426 to L.F. and Y.Z.). The funders had no role in study design, data collection and analysis, decision to publish or preparation of the paper.

### Author contributions

Y.Z. conceived the project and designed the experiment. L.F. gave additional advice on experimental design. W.Z., Q.W. and C.Z. developed the pipeline. W.Z. and Q.W. performed the experiments and

analyzed the data. Y.L. developed machine-learning methods. W.Z., Q.W., L.F. and Y.Z. wrote the paper. All authors proofread and approved the final paper.

## Competing interests

The authors declare no competing interests.

## Additional information

**Correspondence and requests for materials** should be addressed to Lydia Freddolino or Yang Zhang.

# Reporting Summary

## Statistics

For all statistical analyses, confirm that the following items are present in the figure legend, table legend, main text, or Methods section.

| n/a | Confirmed | |
|---|---|---|
| ☐ | ☒ | The exact sample size (*n*) for each experimental group/condition, given as a discrete number and unit of measurement |
| ☐ | ☒ | A statement on whether measurements were taken from distinct samples or whether the same sample was measured repeatedly |
| ☐ | ☒ | The statistical test(s) used AND whether they are one- or two-sided<br>*Only common tests should be described solely by name; describe more complex techniques in the Methods section.* |
| ☒ | ☐ | A description of all covariates tested |
| ☐ | ☒ | A description of any assumptions or corrections, such as tests of normality and adjustment for multiple comparisons |
| ☐ | ☒ | A full description of the statistical parameters including central tendency (e.g. means) or other basic estimates (e.g. regression coefficient) AND variation (e.g. standard deviation) or associated estimates of uncertainty (e.g. confidence intervals) |
| ☐ | ☒ | For null hypothesis testing, the test statistic (e.g. *F*, *t*, *r*) with confidence intervals, effect sizes, degrees of freedom and *P* value noted<br>*Give P values as exact values whenever suitable.* |
| ☒ | ☐ | For Bayesian analysis, information on the choice of priors and Markov chain Monte Carlo settings |
| ☒ | ☐ | For hierarchical and complex designs, identification of the appropriate level for tests and full reporting of outcomes |
| ☒ | ☐ | Estimates of effect sizes (e.g. Cohen's *d*, Pearson's *r*), indicating how they were calculated |

*Our web collection on statistics for biologists contains articles on many of the points above.*

## Software and code

Policy information about availability of computer code

| Data collection | No software was used to collect benchmark dataset. All data are free available on CASP13, CASP14 and CASP15 sub-sections at https://www.predictioncenter.org/index.cgi. MEGAHIT (1.0), Prodigal (2.6), CD-HIT (4.6), FragGeneScan (1.2), and MMseqs2 (version aa175d63658d9aa2e908325a6fd40e9dbb260c9a) were used to collect TaraDB, MetaSourceDB and JGIclust metagenomics databases. |
|---|---|
| Data analysis | The MSAs and structure models were generated by DeepMSA2 (2.0) server (https://zhanggroup.org/DeepMSA2/) and DMFold server (https://zhanggroup.org/DMFold/), and all statistical analyses were done by R (4.1.2) software. AlphaFold2 (2.2.0) was used as control method for checking the modeling quality of protein monomer and multimer. DeepPotential (1.0) was used to generate contact and distance restraints for CASP13-15 protein monomers. HHblits (2.0.15 and 3.1.0), HMMER (3.1b2), BLAST (2.2.26), kClust (1.0), Clustal Omega (1.2.4), and AlphaFold2 (2.2.0) has been used to generate and rank MSAs in DeepMSA2 method. US-align (Version 20220626) was used to analysis the model quality. |

For manuscripts utilizing custom algorithms or software that are central to the research but not yet described in published literature, software must be made available to editors and reviewers. We strongly encourage code deposition in a community repository (e.g. GitHub). See the Nature Portfolio guidelines for submitting code & software for further information.

# Data

Policy information about availability of data

All manuscripts must include a data availability statement. This statement should provide the following information, where applicable:

- Accession codes, unique identifiers, or web links for publicly available datasets
- A description of any restrictions on data availability
- For clinical datasets or third party data, please ensure that the statement adheres to our policy

Third-party databases, Uniclust30 (UniRef30), Uniref90, Metaclust, BFD, MGnify are used this work, those databases are available at https://gwdu111.gwdg.de/~compbiol/uniclust/, https://ftp.uniprot.org/pub/databases/uniprot/current_release/uniref/uniref90/, https://metaclust.mmseqs.org/, https://bfd.mmseqs.com/, and http://ftp.ebi.ac.uk/pub/databases/metagenomics/peptide_database/, respectively. All CASP benchmark data used in this work are available at https://zhanggroup.org/DMFold/ or https://zenodo.org/record/8371924 (https://doi.org/10.5281/zenodo.8371924). The structure modeling results on 5,042 human proteome proteins are freely available at https://zhanggroup.org/DMFold/human for academic use. Source data are provided with this paper.

# Human research participants

Policy information about studies involving human research participants and Sex and Gender in Research.

| | |
|---|---|
| Reporting on sex and gender | NA |
| Population characteristics | NA |
| Recruitment | NA |
| Ethics oversight | NA |

Note that full information on the approval of the study protocol must also be provided in the manuscript.

# Field-specific reporting

Please select the one below that is the best fit for your research. If you are not sure, read the appropriate sections before making your selection.

☒ Life sciences        ☐ Behavioural & social sciences        ☐ Ecological, evolutionary & environmental sciences

For a reference copy of the document with all sections, see nature.com/documents/nr-reporting-summary-flat.pdf

# Life sciences study design

All studies must disclose on these points even when the disclosure is negative.

| | |
|---|---|
| Sample size | The manuscript includes an evaluation of structure predictions for 293 protein monomer targets from CASP13, CASP14 and CASP15, and 54 protein complex targets from CASP13 and CASP14. 38 protein complex targets from the CASP15 blind test are also used in the manuscript. 5,042 human proteins where AlphaFold2 has bad-quality models are modeled by DMFold, and the query sequences are taken from UniProt. No statistical method was used to decide the sample size, but the number of the samples is sufficient for applied statistical analysis in each case. In addition, the CASP datasets were taken from the community-wide experiments, where all datasets are standard testing sets in this field. |
| Data exclusions | The proteins homologous (based on the release date) to the benchmark dataset were excluded from the template library to avoid homologous contamination. Six CASP domains are excluded from threading benchmark test because HHsearch failed to generate results with some MSAs from third-party control methods (i.e., HMMER MSAs contain too many sequences). 22 CASP domains are excluded from DeepPotential restraint prediction benchmark test because DeepPotential failed to generate results with some MSAs from third-party control methods (i.e., HMMER MSAs contain too many sequences), or some targets are extremely hard for DeepPotential to make a prediction (i.e., MAE>30 for all MSAs). Based on simple statistic analysis (i.e., average value), excluding those data dose not change the conclusion that DeepMSA2 performs better than the five control methods. |
| Replication | All results could be reproduced by our server and standalone package with the full version databases, or based on the information provided in manuscript. All experiments are done independently without any technical replication. |
| Randomization | There is no random method used in the manuscript. All data in CASP13-15 are used in the manuscript. Covariants are not relevant in this study since the CASP benchmark dataset is the blind and golden standard benchmark dataset in benchmarking protein structure prediction, and most of the 'FM' targets in CASP datasets do not have homologous structure in PDB, and thus are not redundant with others. |
| Blinding | There was no blinding group or benchmarking analysis on CASP13 and CASP14 datasets in this manuscript. The CAPS15 results are taken from CASP official website, and the authors participated the CASP15 world-wide experiment test with blinding, where all experimental structures |

were not available to the authors during the CASP15 season; thus, results from CASP15 represent a community-wide standard blind experiment.

# Reporting for specific materials, systems and methods

We require information from authors about some types of materials, experimental systems and methods used in many studies. Here, indicate whether each material, system or method listed is relevant to your study. If you are not sure if a list item applies to your research, read the appropriate section before selecting a response.

## Materials & experimental systems

| n/a | Involved in the study |
|-----|----------------------|
| ☒ ☐ | Antibodies |
| ☒ ☐ | Eukaryotic cell lines |
| ☒ ☐ | Palaeontology and archaeology |
| ☒ ☐ | Animals and other organisms |
| ☒ ☐ | Clinical data |
| ☒ ☐ | Dual use research of concern |

## Methods

| n/a | Involved in the study |
|-----|----------------------|
| ☒ ☐ | ChIP-seq |
| ☒ ☐ | Flow cytometry |
| ☒ ☐ | MRI-based neuroimaging |

