## [Peer Review File · Nature Methods]

Peer Review Information

Manuscript Title: Improving deep learning protein monomer and complex structure prediction using DeepMSA2 with huge metagenomics data

Corresponding author name(s): P. Lydia Freddolino, Yang Zhang

Editorial Notes: None

Reviewer Comments & Decisions:

Decision Letter, initial version:

Dear Dr. Zhang,

Your Article, "Improving deep learning protein monomer and complex structure prediction using DeepMSA2 with huge metagenomics data", has now been seen by 2 reviewers. As you will see from their comments below, although the reviewers find your work of considerable potential interest, they have raised a number of concerns. We are interested in the possibility of publishing your paper in Nature Methods, but would like to consider your response to these concerns before we reach a final decision on publication.

We therefore invite you to revise your manuscript to address these concerns. We would like to see more suitable and fairer comparisons between the tested methods, as recommended by reviewer #1. Additionally, as both reviewers mention, some of the claims related to the method and performance should be suitably toned down.

* include a point-by-point response to the reviewers and to any editorial suggestions

* please underline/highlight any additions to the text or areas with other significant changes to facilitate review of the revised manuscript

* address the points listed described below to conform to our open science requirements

* ensure it complies with our general format requirements as set out in our guide to authors at www.nature.com/naturemethods

* resubmit all the necessary files electronically by using the link below to access your home page

[Redacted] This URL links to your confidential home page and associated information about manuscripts you may have submitted, or that you are reviewing for us. If you wish to forward this email to co-authors, please delete the link to your homepage.

We hope to receive your revised paper within 8 weeks. If you cannot send it within this time, please let us know. In this event, we will still be happy to reconsider your paper at a later date so long as nothing similar has been accepted for publication at Nature Methods or published elsewhere.

OPEN SCIENCE REQUIREMENTS

REPORTING SUMMARY AND EDITORIAL POLICY CHECKLISTS

Please note that these forms are dynamic ‘smart pdfs’ and must therefore be downloaded and completed in Adobe Reader. We will then flatten them for ease of use by the reviewers. If you would like to reference the guidance text as you complete the template, please access these flattened versions at <http://www.nature.com/authors/policies/availability.html>.

DATA AVAILABILITY

All novel DNA and RNA sequencing data, protein sequences, genetic polymorphisms, linked genotype and phenotype data, gene expression data, macromolecular structures, and proteomics data must be deposited in a publicly accessible database, and accession codes and associated hyperlinks must be provided in the “Data Availability” section.

Please include a “Data availability” subsection in the Online Methods. This section should inform readers about the availability of the data used to support the conclusions of your study, including accession codes to public repositories, references to source data that may be published alongside the paper, unique identifiers such as URLs to data repository entries, or data set DOIs, and any other statement about data availability. At a minimum, you should include the following statement: “The data that support the findings of this study are available from the corresponding author upon request”, describing which data is available upon request and mentioning any restrictions on availability. If DOIs are provided, please include these in the Reference list (authors, title, publisher (repository name),

identifier, year). For more guidance on how to write this section please see:

<http://www.nature.com/authors/policies/data/data-availability-statements-data-citations.pdf>

CODE AVAILABILITY

Please include a “Code Availability” subsection in the Online Methods which details how your custom code is made available. Only in rare cases (where code is not central to the main conclusions of the paper) is the statement “available upon request” allowed (and reasons should be specified).

MATERIALS AVAILABILITY

SUPPLEMENTARY PROTOCOL

To help facilitate reproducibility and uptake of your method, we ask you to prepare a step-by-step Supplementary Protocol for the method described in this paper. We [encourage authors to share their step-by-step experimental protocols](https://www.nature.com/nature-research/editorial-policies/reporting-standards#protocols) on a protocol sharing platform of their choice and report the protocol DOI in the reference list. Nature Portfolio's Protocol Exchange is a free-to-use and open resource for protocols; protocols deposited in Protocol Exchange are citable and can be linked from the published article. More details can found at <https://www.nature.com/nature-research/editorial-policies/reporting-standards#protocols>

href="https://www.nature.com/protocolexchange/about"
target="new">www.nature.com/protocolexchange/about.

ORCID

Sincerely,
Arunima

Arunima Singh, Ph.D.
Senior Editor
Nature Methods

Reviewers' Comments:

Reviewer #1:

Remarks to the Author:

Summary:

AlphaFold have significantly improved the accuracy of protein structure prediction. However, it critically depends on having a large and sufficiently diverse set of multiply aligned homologous proteins, not readily available for all queries. AlphaFold performance in predicting the structure of protein complexes is even harder, in particular for hetero-oligomers. This is mostly because construction of multiple sequence alignment (MSA) requires proper linking of the relevant homologues for each of the different

monomers in the complex. Zhang and coworkers introduce DeepMSA2, an advanced pipeline for improving MSA construction, and a related pipeline DMFold that uses this improved MSA and the AlphaFold algorithm for structure prediction of protein monomers and complexes.

Opinion:

According to the manuscript MSAs constructed using DeepMSA2 are deeper and more balanced compare to the leading alternatives, and DMFold produce model structures that are even more accurate than those of AlphaFold. I listed a few issues to be addressed in order to convince me that this is really the case. For now, I am convinced about the improvement with protein complexes (Figure 5) but not necessarily with monomers.

Major issues:

1. "Overall, DeepMSA2 demonstrates a balanced ability to detect diverse homologous sequences with a high alignment coverage, which likely contributes to the improved ability of DeepMSA2 MSAs to support protein structure prediction": This sentence summarizes the results presented in Figure 2. However, my own interpretation of that data is that DeepMSA2 is comparable to HHblitz and MMsecs2. To my surprise even the good old PSI-BLAST is roughly in this range. Taking into account that these methods were examined on different settings in terms of databases, etc, to the settings they were designed with, i.e., presumably sub-optimal, the differences appear marginal.
2. Table S2. Indeed, DeepMSA2 managed to find a template with better TM-score compared to the rest of the MSAs. However, the score itself is only slightly better than the next one (by HHblitz). (And I do not see the p-values that the table legend refers to.) On the same subject, Fig. 2D that is argued to show better template selection with DeepMSA2 compared to HHsearch, does not support this statement. Or maybe I do not know how to view it properly.
3. "In Fig 2F, we further display the mean absolute distance error (MAE, see Eq 5) of the top 5L long-range distance map prediction by DeepPotential, where the use of the MSA from DeepMSA2 results in an MAE=2.41Å that is significantly lower than those from the other five MSA programs, i.e., 3.32Å for BLAST, 2.78Å for PSIBLAST, 2.73Å for MMseqs2, 2.78Å for HHblits, and 3.03Å for HMMER, all with a p-value<0.05 by one-sided Student's t-test": Again, to me the distributions look very similar. I wonder how the minor differences are statistically significant after all.
4. "Thus, thanks to the balanced, high-information MSAs that it provides, DeepMSA2 can help guide more accurate template recognition and spatial restraint predictions, which are critical for the prediction of protein tertiary structure": Again, I disagree with this summary statement. Maybe there is a bit of improvement, but it is anticipated, given the sub-optimal setting for the competitor's MSAs.
5. Figure 3A. The vast majority of datapoints are in the upper right corner. They appear right along the diagonal, which would mean that DMFold is as good as AlphaFold. The main text says that DMFold is

better, I guess this statement refers to the 20 or so other datapoints, most of which are above the diagonal. I'd revise the text to explain that. Same comment also regarding Fig. 5A.

6. "Since the major difference between DMFold and AlphaFold2 is in MSA generation, the large improvement in TM-score indicates that the quality of the MSAs has a profound impact on the structural models that are ultimately generated.": This sentence refers to the minor improvements presented in Fig. 3. It should be tuned down.

7. Fig. S2: In the upper left example the DMFold and AlphaFold models look very similar to each other (and to the experimental structure). Maybe TM-score difference of 0.1 is not high enough to show diversity? Maybe try 0.2? And, of course, tune down the statement about improvements over AlphaFold?

8. Figure 4 convincingly shows that in spite of my reservations above DeepMSA2 managed to produce significantly better MSAs that enabled accurate prediction of nearly 2000 human proteins for which AlphaFold models were not trustworthy. To further understand this apparent conflict, I explored a bit further. Indeed, the examples shown in the manuscript are very convincing. However, I clicked around in the database of ~5000 human proteins and in most cases the protein core, as predicted by DeepMSA2, seemed very similar to that of the AlphaFold model. I encourage the authors to highlight all the ~2000 models that they consider better than the corresponding AlphaFold models.

9. Further on this: Having the per-residue measures that AlphaFold provide may have helped convincing me to the contrary.

10. I encourage the authors to include the per-residue pLDDT score in the database because it provides also local indications on structure quality, allowing to figure out which parts of the structure are more reliable.

11. And I am also missing the predicted aligned errors (PAE), i.e., the expected positional error at residue x if the predicted and actual structures are aligned on residue y .

12. Figure 6. Since the structures are known it is trivial to superimpose them on the models. Thus, it makes sense to measure RMSD, which more accurately shows similarity.

13. Methodology: Just to clarify, AlphaFold was retrained within the context of the improved MSA pipeline, right? Or maybe used as is?

14. When predicting the structure of a multimer, I take it that in the current implementation the stoichiometry is provided. I wonder however if it is possible to deduce it from the coevolution signal. In particular for homo-oligomers.

Reviewer #2:

Remarks to the Author:

Two main messages of the paper are that deeper and better constructed MSAs improve protein structure prediction and that this enhancement can push model accuracy beyond the 'vanilla' AlphaFold2 version (both standard and multimer), in many cases generating much better models. The

method was tested in CASP15 and was recognized as one of the best performers. The study is well thought-through, the message is appropriately substantiated with the data, examples are illustrative and well picked, and the paper is overall well written.

Comments below.

ABSTRACT

'An integrated pipeline with DeepMSA2 participated in the most recent CASP15 experiment and created high quality complex models with mean Z-scores 3-times higher than the AlphaFold2-Multimer server.'

Even though technically correct, this statement may mislead readers to think that the reported method is 3 times better than AF2-Multimer. First and foremost, it is not an apple-to-apple comparison as the databases are different. I guess that if the available AF2-M method was simply retrained with the 'huge metagenomics data' (without any other changes), it would show comparable performance. This was demonstrated on monomeric CASP15 targets with an updated AF2 version 2.3.0 released in the end of 2022. I suggest the authors tone down this statement (here and in the Discussion).

INTRODUCTION

lines 57-59: These lines are formulated cautiously (which is good), however an unexperienced reader may still downplay a very important word 'some' in the phrase 'on some orphan sequences for which detectable homologous sequences do not exist'. While in the next paragraph the authors do emphasize that language models in CASP15 were in fact inferior to the state-of-the-art methods, it is not clear whether the methods mentioned here are among those discussed further on. As an easy fix, I would suggest substituting 'proved to be capable of generating' with 'can generate'.

62-65: Indeed, in CASP14, results in assembly prediction were not as excited as those in tertiary structure prediction. However, this changed in CASP15, and the authors are surely aware of it and should adjust their text accordingly.

METHODS

485-488: CASP assessment is usually done on domains (evaluation units), and not targets (some targets can contain domains of different homology-based categories). Please clarify that the numbers provided here (and in figures) are for domains. Also, typically CASP domain classification includes an overlap TBM/FM difficulty category. No mentioning of that here. Checking Table S9 I can see that TBM/FM domains were included as a subclass of a broader FM category. Please clarify in the text.

486-487: 'the effect of DeepMSA2 on monomer protein structure folding'. Protein folding does not depend on MSAs. It has its own rules and pathways, often unknown. However, it does seem like modern-day protein structure prediction depends on MSAs.

498-505: Do not understand arithmetic here. $13,838 + 6,757 = 20,595$. This is more than 20,389 proteins mentioned as the Human Proteome dataset in your paper (including the 2-residue peptide). Where does the discrepancy come from?

507-513: There are many different Neff definitions and protocols to calculate them. It would be nice if the authors can help their readers to understand the Neff definition provided here. Practically all Neffs in the main text are discussed from the comparative analysis perspective, i.e., this method recovers more diverse sequences than the other, and thus is better. But what about the absolute values? What Neff values according to the provided definition indicate a diverse enough MSA and a useful for the prediction purposes evolutionary signal? 1, 2, 5, 10? Or in other words, how many diverse sequences (below 0.8 cutoff) are needed in a MSA for, say, 100- or 300-residue long protein? What about the coverage of the query sequence?

Figure 6: recommend swapping names of panels (A <-> B) in the figure and the caption. It was confusing for the reviewer that the upper left panel in the Figure was B and not A.

773-775: Do not like 3 things: 1) the 'DMFold-Multimer' typo, 2) text in the 1st parenthesis, and 3) the fact that it is not clarified that the AF2-Multimer results are from the March 2022 edition of a public server. Suggest correcting to: (B) Sum of Z-scores on 43 multimeric targets for 87 CASP15 assembly groups. DMFold-Multimer (registered as 'Zheng') and the public March-2022 version of the AlphaFold2-Multimer server (registered as 'NBIS-AF2-multimer') are marked in red and yellow, correspondingly.

RESULTS

96-97: 'The optimal multimer MSAs are then selected based on a combined score of the depth of the MSAs and folding score of the monomer chains'. Did not see it in the Methods where the 'folding score' of monomer chains is defined or used.

98: There is no 'Methods' section (line 482 lists 'ONLINE METHODS', adjust either here or there).

133: Well, TM=0.483 is not considerably higher than 0.469 or 0.477. They are all rather very similar. So nothing to brag about here.

135, 140, Figure 2: What does the 'top L long-range contacts' mean? Is L defined anywhere that I missed? And the phrase 'top 5L long-range distance map' is simply confusing.

138-139: What are the numbers in parentheses defining accuracy of a contact map?

139: 'A similar tendency can be seen FOR short- and ...'

135-146: When I look at graphs in Fig. 2 (in particular, E and F) I cannot see anything SIGNIFICANTLY better than the rest (as the authors continuously emphasize). In the graphs, BLAST is always provided second after the DeepMSA2 and it gives an impression of a sizeable improvement. But BLAST results are expected to be poorer (as BLAST is a conceptually different and simpler method), and if one removes BLAST's violin in the lineup of methods (or moves it to the very right), then the 'significantly better' impression disappears. It is better to move to a quantitative language in the text.

152: There were no that many FM targets in CASP13 + CASP14. I guess that the 85 FM mentioned in the text is for a wider target set including slightly easier FM/TBM domains.

188-191: Please reformulate - hard to read.

198-199: have a confidence score (pLDDT<0.7) -> have a confidence score of pLDDT<0.7.

204: Please explain also in the Figure 4 caption what is the '1,934' number shown there (it points in the y-axis direction, which shows pLDDT). Also, explain (in addition to the main text) that the '5042 difficult targets' mentioned there are those specifically selected by you where AF2's pLDDT was <0.7, as by quickly looking at the histogram it may seem that AF2 is unable to generate models in excess of 0.7. In general, I think that the panel A may be confusing for a reader, and I would recommend (but not insist) to delete the AF2 histogram as it delivers no useful message in itself in the discussed context.

195-210:

1) Have any predicted structures from the human proteome been solved experimentally recently? Especially from the 5,042? It would be interesting to compare the results to known answers.

2) Also, it is nice that in 1,943 cases the pLDDT score jumped over 0.7. However, I am curious, in how many cases these are true positives (i.e., represent better predictive ability of DMFold) compared to

false positives (e.g., the target is largely disordered and low AF2's pLDDT scores are genuinely indicating that). But I guess it is impossible to check this without the experimental structural data, or?

296-297: content: 'where the former two measure the global fold quantity' [?]
and grammar: and the latter two for assessing protein interface modeling quality of protein complexes.

297-298: 'The standard version of AlphaFold2-Multimer also participated in the CASP15 with registered name of 'NBIS-AF2-multimer' as operated by the Elofsson lab.'

Despite your explanations, it still sounds like DeepMind participated in CASP15, which is not true.

I guess the sentence in question can be deleted and then you can say:

Overall, DMFold-Multimer achieved a cumulative Z-score of 35.43, which is nearly 3 times higher than that of the 'NBIS-AF2-multimer' group (i.e., the public AlphaFold2-Multimer server run by the Elofsson Lab on CASP15 targets) (12.30), and 18.3% higher than the second-best performing group (29.95).

299-300: Please make sure your numbers are updated to reflect those provided at the link in line 293.

301-310: Talking about the immune complexes. Can you explain why these three were modeled very well, while others that are similar (e.g., H1142) were not?

A QUESTION related to the subject of the paper, but not directly to the material discussed in the paper: The paper proves that deep learning methods can generate good structure models when evolutionary information is abundant (deep MSAs). How do the authors see perspectives of deep learning methods for the RNA structure prediction, where structural data are much sparser?

Noticed grammar issues: lines 95, 516-517, 538, 595, 645-646, 691-692, 185, 235-236

Author Rebuttal to Initial comments

Response to Editor

We very much appreciate the suggestions from the Editor, which are very helpful to guide our manuscript revision. More specifically, the Editor suggested:

We therefore invite you to revise your manuscript to address these concerns. We would like to see more suitable and fairer comparisons between the tested methods, as recommended by reviewer #1. Additionally, as both reviewers mention, some of the claims related to the method and performance should be suitably toned down.

We appreciate the opportunity the Editor gave us to revise our manuscript and address the Reviewers' concerns.

One of the major concerns of Reviewer #1 is on the relative marginal improvement of DeepMSA2 on some of the MSA characteristics compared to other methods. We found that some of the presented parameters (*Neff*, *SeqID*, *cov*) in our previous comparisons do not closely reflect the ability of MSAs for extracting essential characteristics of evolutionary and co-evolutionary information, and therefore reorganized the manuscript by focusing discussions on the qualities which are more closely associated to its ability to assist deep-learning protein structure prediction, where consistent and significant improvements were found in all these qualities in the enlarged testing dataset (see our Response to Point 3 of Reviewer #1).

Meanwhile, we included a new set of 94 protein domains from CASP15 in our test dataset to enhance the statistics of our benchmark tests. Especially because the structures of these proteins were solved after the AlphaFold2 was released during CASP14 and can therefore avoid over-training of the algorithms, the inclusion of these new protein targets helps to facilitate a more objective comparison between AlphaFold2 and our pipeline (see our Response to Point 4 of Reviewer #1)

Furthermore, to address the concern of Reviewer #2 on the lack of experimental validation of the DMFold models on the human proteome structures, we have identified 48 newly solved structures of proteins from this set, and carefully examined the model accuracy relative to the experimental structures. The results demonstrated additional evidence on the importance of effective MSA constructions by DeepMSA2 on high-accuracy structure predictions of unknown proteins (see our Response to Point 21 of Reviewer #2).

Finally, we have toned down some of the potentially aggressive statements related to our method and performance, following the suggestion of the Editor and the Reviewers (see our Responses to Point 5 and 6 of Reviewer #1 and Point 2 of Reviewer #2).

Response to Reviewer #1

We very much appreciate the comments and suggestions from the Reviewer, which we found very helpful for improving the quality of both the manuscript and the online servers. One of the major concerns of the Reviewer is the relative marginal improvement of DeepMSA2 on some of the MSA characteristics compared to other methods. We found that some of the presented parameters (*Neff*, *SeqID*, *cov*) do not closely reflect the ability of MSAs for extracting essential characteristics of evolutionary and co-evolutionary information, and therefore reorganized the manuscript by focusing discussions on the qualities which are more closely associated to its ability to assist deep-learning protein structure prediction rather than on superficial parameters that do not directly contribute to the structure prediction aim. The second major concern of the Reviewer is on the improvement of DMFold on monomer proteins. To address this issue, we have added additional tests on the new CASP15 targets, which reinforced the statistics and demonstrated significant advancements of DMFold over leading alternatives in multiple aspects of structure prediction. Additionally, we made significant changes on the manuscript to carefully address other concerns of the Reviewer on the results of human proteome structure prediction, webserver library, confidence score presentation (per-residue pLDDT and *PAE*), and stoichiometry of complex structure prediction. We have also turn down several statements according to the Reviewer's suggestion. In the text below, we include point-by-point replies to the comments of the Reviewer, where all changes have been highlighted in yellow in the manuscript.

1. The Reviewer commented:

Summary:

AlphaFold have significantly improved the accuracy of protein structure prediction. However, it critically depends on having a large and sufficiently diverse set of multiply aligned homologous proteins, not readily available for all queries. AlphaFold performance in predicting the structure of protein complexes is even harder, in particular for hetero-oligomers. This is mostly because construction of multiple sequence alignment (MSA) requires proper linking of the relevant homologues for each of the different monomers in the complex. Zhang and coworkers introduce DeepMSA2, an advanced pipeline for improving MSA construction, and a related pipeline DMFold that uses this improved MSA and the AlphaFold algorithm for structure prediction of protein monomers and complexes.

We appreciate the nice summary and positive comments from the Reviewer on the work.

2. The Reviewer commented:

Opinion:

According to the manuscript MSAs constructed using DeepMSA2 are deeper and more balanced compare to the leading alternatives, and DMFold produce model structures that are even more accurate than those of AlphaFold. I listed a few issues to be addressed in order to convince me that this is really the case. For now, I am convinced about the improvement with protein complexes (Figure 5) but not necessarily with monomers.

Thank you for the comments. We are happy to know that the Reviewer is convinced that DMFold outperforms AlphaFold2, one of the leading approaches of the field, in protein complex structure prediction. In fact, the advancements of DMFold over AlphaFold2 in both monomer and complex

structure predictions are equally significant based on both benchmarking and blind CASP test results. Specifically for monomers, our analysis was previously based on CASP13 and CASP14 targets. Since AlphaFold2 was developed during CASP14, some of these targets may have been well-optimized by AlphaFold2. Thus, we added a new test set of proteins from the CASP15 experiment in the revised manuscript, which were released after the AlphaFold2 training. The results showed again the advancement of our pipeline over AlphaFold2 on the new monomer benchmark dataset. More detailed discussion on the monomer protein structure prediction of DMFold and AlphaFold2 are explained below (see Points 4, 6, 7, and 8).

3. The Reviewer commented:

Major issues:

1. *“Overall, DeepMSA2 demonstrates a balanced ability to detect diverse homologous sequences with a high alignment coverage, which likely contributes to the improved ability of DeepMSA2 MSAs to support protein structure prediction”: This sentence summarizes the results presented in Figure 2. However, my own interpretation of that data is that DeepMSA2 is comparable to HHblitz and MMseqs2. To my surprise even the good old PSI-BLAST is roughly in this range. Taking into account that these methods were examined on different settings in terms of databases, etc, to the settings they were designed with, i.e., presumably sub-optimal, the differences appear marginal.*

2. *Table S2. Indeed, DeepMSA2 managed to find a template with better TM-score compared to the rest of the MSAs. However, the score itself is only slightly better than the next one (by HHblitz). (And I do not see the p-values that the table legend refers to.) On the same subject, Fig. 2D that is argued to show better template selection with DeepMSA2 compared to HHsearch, does not support this statement. Or maybe I do not know how to view it properly.*

3. *“In Fig 2F, we further display the mean absolute distance error (MAE, see Eq 5) of the top 5L long-range distance map prediction by DeepPotential, where the use of the MSA from DeepMSA2 results in an MAE=2.41Å that is significantly lower than those from the other five MSA programs, i.e., 3.32Å for BLAST, 2.78Å for PSIBLAST, 2.73Å for MMseqs2, 2.78Å for HHblits, and 3.03Å for HMMER, all with a p-value<0.05 by one-sided Student’s t-test”: Again, to me the distributions look very similar. I wonder how the minor differences are statistically significant after all.*

4. *“Thus, thanks to the balanced, high-information MSAs that it provides, DeepMSA2 can help guide more accurate template recognition and spatial restraint predictions, which are critical for the prediction of protein tertiary structure”: Again, I disagree with this summary statement. Maybe there is a bit of improvement, but it is anticipated, given the sub-optimal setting for the competitor’s MSAs.*

We thank the Reviewer for the comments, which raised several important issues on the significance of the improvement brought by DeepMSA2 compared to the control methods; these concerns are all related to the data in **Figure 2**.

First, we would like to point out that the former comparison data in the original **Figures 2A-2C** on the number of effective sequence (*N_{eff}*), sequence identity (*SeqId*) and alignment coverage (*cov*), on which DeepMSA2 had the output value in the middle of the control methods, can be misleading, because these parameters only account for a few geometrical aspects of the MSAs and do not necessarily reflect their ability to encode evolutionary/co-evolutionary features that are

critical to deep-learning model training. In fact, many of the studies have shown a weak (or none) correlation between the accuracy of deep-learning models and *Neff* (or *SeqId* and *cov*) of MSAs (see, e.g., Jumper et al, Nature 696: 583, 2021; Li et al, Bioinformatics, 35: 4647, 2019; Buchan et al, Proteins, 86:78, 2018). To avoid confusion, we have moved the original **Figures 2A-2C** of (*Neff*, *SeqId* and *cov*) to the Supplementary Information. Accordingly, we only kept a brief discussion of these parameters, given that these discussions can help us to understand the general shape and overall balance property of MSA construction process of different programs.

Second and most importantly, we have the discussions in the Section focused more on the TM-score of template recognitions and accuracy of contact/distance map predictions, which are more directly related to the evolutionary and co-evolutionary information of MSAs. Following the Reviewer's suggestion and to assess the statistical significance of the comparisons more quantitatively, we added *P*-values to **Tables S2-4** comparing DeepMSA2 and the control methods. The results showed that the *P*-values are below 0.05 in all the categories of comparisons, including both FM and TBM domains. If we count for all the 293 domains from CASP13-15, the *P*-values are all below $1.0E-07$, showing that the improvements brought by DeepMSA2 are indeed highly statistically significant. The reason for the high significance *P*-values is because the improvements are consistent, i.e., there are far more targets in which DeepMSA2 achieved an improvement than the targets in which DeepMSA2 did not (see, e.g., the head-to-head comparison in **Figure S3**), although the difference on average may not be very large. Meanwhile, as mentioned above we have included more targets from CASP15, which help increase number of the analyses and stability of the statistics.

Figure S3. Head-to-head comparisons of top 5L long-range mean absolute distance error (*MAE*) between DeepMSA2 and (A) BLAST, (B) PSIBLAST, (C) MMseqs2, (D) HHblits, and (E) HMMER on 287 monomer protein domains from CASP13-15. Points below the diagonal indicate better performance by DeepMSA2 relative to each control. This analysis has excluded six chains from a protein complex H1137, which form an interwound alpha-helix barrel (see **Figure 6B**) and therefore the contact/distance maps for each of the long alpha-helix chains are irrelevant for DeepPotential predictions.

Furthermore, we have redrawn **Figure 2** and converted the violin plots to bar-plots, which help better highlight the performance difference between different methods. Again, the data shows that DeepMSA2 outperforms the control methods consistently in all categories of targets on the critical task of assisting template recognition and deep-learning contact/distance map predictions.

Accordingly, we have rewritten the discussions to clarify the issues (Page 3–4):

Nevertheless, the parameters considered above (N_{eff} , $SeqId$, cov) only measure the geometrical characters of the MSA matrix, and do not necessarily reflect the inherent evolutionary and co-evolutionary information contained in the MSAs (**Fig S2**), which are critical to the deep learning-based protein structure predictions. As a more direct test of their ability to encode evolutionary and co-evolutionary information, we further examine the performance of the MSAs in assisting template recognition and deep learning spatial restraint prediction. In **Fig 2A**, we list the average TM-scores of the structure templates recognized by HHsearch based on the profile HMMs constructed from the six different MSAs, where all close homologous templates (>30% sequence identity to the query sequence) have been excluded from the template library. It is shown that the templates detected using the DeepMSA2 MSA have the highest TM-score for both FM and TBM domains. The average TM-score for all 293 CASP domains obtained using the MSA from DeepMSA2 (0.492) is also higher than those using the MSAs from BLAST (0.454), MMseqs2 (0.469), HHblits (0.463), HMMER (0.448), or PSIBLAST (0.448), all with p -values < 2.72E-08 by one-sided Student's t -test (**Table S2**).

In **Fig 2B**, we present the precision of the top L long-range contact predictions made by the deep neural-network program DeepPotential^{11, 23}, using co-evolutionary features derived from the six different MSAs (where L is the query sequence length, and “long-range” represents a sequence separation $|i - j| \geq 24$ residues for the contacts between residues i and j , which are then ranked by the DeepPotential contact probability). Again, utilizing the DeepMSA2 MSA results in a higher precision of top L long-range contacts (=0.601) predicted by DeepPotential, compared to those obtained while using the MSAs from BLAST (=0.514), MMseqs2 (0.568), HHblits (0.559), HMMER (0.538), or PSIBLAST (0.566) as inputs for DeepPotential. A similar tendency can be seen for top $L/5$ and $L/2$ predictions as detailed in **Table S3**, where p -values are below 1.23E-05 for all the comparisons when all domains in our evaluation set are considered.

In **Fig 2C**, we further display the mean absolute distance error (MAE , see Eq. 6) of the top $5L$ long-range distances predicted by DeepPotential, where the use of the MSA from DeepMSA2 results in an $MAE=2.22\text{\AA}$, which is significantly lower than those from the other five MSA programs, i.e., 3.09\AA (p -value=1.62E-22) for BLAST, 2.70\AA (p -value=5.79E-12) for PSIBLAST, 2.68\AA (p -value=1.03E-07) for MMseqs2, 2.74\AA (p -value=1.26E-09) for HHblits, and 2.98\AA (p -value=4.80E-13) for HMMER (**Table S4**). In **Fig S3**, we also display a head-to-head comparison of the MAE between DeepMSA2 and the five control methods, where DeepMSA2 has lower MAE values than the other MSA methods for a dominant fraction of the domains; this accounts for the major reason for the significant p -values. Overall, these data show that the balanced and highly informative MSA construction provided by DeepMSA2 might have encoded more relevant co-evolutionary features and help guide accurate template recognition and spatial restraint predictions; this ability is also important for the subsequent deep learning-based tertiary structure prediction.

Finally, we note that despite the improved performance by DeepMSA2 on template recognition and contact/distance map prediction, the data presented in this Section and **Figure 2** are only for pilot examination of the method. The more critical examinations of DeepMSA2 on tertiary and quaternary structure predictions, which are the goals of DeepMSA2 development, are presented in the subsequent sections as discussed below.

4. The Reviewer commented:

5. Figure 3A. The vast majority of datapoints are in the upper right corner. They appear right along the diagonal, which would mean that DMFold is as good as AlphaFold. The main text says that DMFold is better, I guess this statement refers to the 20 or so other datapoints, most of which are above the diagonal. I'd revise the text to explain that. Same comment also regarding Fig. 5A.

Thank you for the comment and suggestion. To address the issue, we first included more targets from CASP15 which increased the statistics of analyses and therefore better highlighted the differences between the methods. As mentioned above, since CASP15 domains were released after AlphaFold2 model training, the comparison results can be less affected by the danger of over-training.

Second, we have split the comparisons in two categories. As shown in **Figure 3A**, for the 86 monomer domains with both TM-scores by AlphaFold2 and DMFold >0.8 , as the Reviewer pointed out, the overall performance is indeed comparable (0.925 vs 0.922 for DMFold and AlphaFold2). For the rest of 46 domains, the TM-score difference for these two methods is highly significant, i.e., average TM-scores of 0.626 vs 0.517 for DMFold and AlphaFold2, with a p -value= $2.86E-04$ in Student's t-test. This highlights the essential point that while AlphaFold2 provides excellent performance on many domains, DMFold is nevertheless capable of providing accurate structures for many of the cases where AlphaFold2 fails. We have rewritten the discussion as following (Page 4):

The DeepMSA2-driven monomer protein prediction pipeline utilizes a modified version of AlphaFold2²⁴, in which the input MSA is replaced with the MSA created by DeepMSA2. In the following discussion, we use 'DMFold' to refer to our pipeline for brevity. In **Fig 3A**, we compare the TM-scores of all models predicted by DMFold vs. AlphaFold2 on the 132 FM monomer proteins from the CASP13-15 experiments. To correctly reflect the FM nature of the domains, all templates released after May 2018, May 2020, and May 2022 have been excluded for the CASP13, CASP14, and CASP15 domains, respectively, when running the programs. It is shown that DMFold generated models with a higher TM-score than AlphaFold2 in 63% (=83/132) of the cases. The average TM-score of the models generated by DMFold (0.821) is 5% higher than that of AlphaFold2 (0.781), with a p -value= $1.82E-04$ in one-sided Student's t-test indicating that the difference is statistically significant. It is notable that the difference mainly comes from difficult domains. For the 86 domains where both AlphaFold2 and DMFold achieved TM-score >0.8 , for example, the average TM-score is very close (i.e., 0.925 vs 0.922 for DMFold and AlphaFold2). But for the remaining 46 domains, where at least one of the methods performed poorly, the TM-score difference is dramatic (i.e., 0.626 for DMFold vs 0.517 for AlphaFold 2), with a p -value= $2.86E-04$ in one-sided Student's t-test. Among the 46 difficult domains, DMFold builds models with TM-scores 0.1 unit higher than AlphaFold in 18 domains while AlphaFold2 does so only in 4 domains. This data highlights the advantage of the DeepMSA2/DMFold pipeline for modeling difficult protein domains.

Similarly, for **Figure 5A**, we split the complex targets into two categories. For the targets where both DMFold- and AlphaFold2-Multimer TM-scores were above 0.9, accounting for 26 targets, the average TM-score is very close (0.961 vs 0.960 for DMFold-Multimer and AlphaFold2-Multimer). However, for the remaining 28 complexes, the average TM-score by DMFold-Multimer (0.716) is 32% higher than that of AlphaFold2-Multimer (0.542), with p -value= $1.05E-04$ in Student's t-test. Thus, again we see that DMFold-Multimer shows its strength on cases where the standard AlphaFold2-multimer cannot provide good predictions. We accordingly added the following discussion in Page 6:

Fig 5A also displays a head-to-head comparison of the TM-score of the models, where DMFold-Multimer outperforms AlphaFold2-Multimer in 70% of cases. Again, the improvement mainly occurs on the difficult complexes: If we consider the 26 easy targets where both DMFold-Multimer and AlphaFold2-Multimer models have TM-scores above 0.9, the average TM-scores are very close (0.961 for DMFold-Multimer vs 0.960 for AlphaFold2-Multimer). For the 28 more difficult targets, however, the average TM-score of DMFold-Multimer (0.716) is significantly higher than that of AlphaFold2-Multimer (0.542), with a p -value=1.05E-04 in one-sided Student's t-test (Fig 5A).

5. The Reviewer commented:

6. "Since the major difference between DMFold and AlphaFold2 is in MSA generation, the large improvement in TM-score indicates that the quality of the MSAs has a profound impact on the structural models that are ultimately generated.": This sentence refers to the minor improvements presented in Fig. 3. It should be tuned down.

Thank you for the suggestion. We have rewritten the sentence to appropriately tone down the statement (Page 4):

Since the major difference between DMFold and AlphaFold2 is in MSA generation, the improvement in TM-score observed for DMFold indicates that the quality of the MSAs has a strong effect on the structural models.

6. The Reviewer commented:

7. Fig. S2: In the upper left example the DMFold and AlphaFold models look very similar to each other (and to the experimental structure). Maybe TM-score difference of 0.1 is not high enough to show diversity? Maybe try 0.2? And, of course, tune down the statement about improvements over AlphaFold?

Thank you for the comment and suggestion. To better highlight the structural improvement, we have redrawn Figure S4 (previously Figure S2) as follows: (1) we introduced 8 examples from CASP13-15 with a TM-score difference above 0.3; (2) we drew the DMFold and AlphaFold2 models separately when they are superposed on the experimental structures, which helps more clearly display the structural differences. Accordingly, we rewrote the following paragraph (Page 5):

In Fig S4, we further list eight other examples from CASP13-15 (T0991-D1, T1064-D1, T1125-D1, T1125-D2, T1125-D5, T1130-D1, T1169-D1, and T1169-D4), in which the TM-score improvements by DMFold are higher than 0.3. In seven out of these eight cases, the N_{eff} of DeepMSA2 is higher than that of AlphaFold2. These results again demonstrate the capacity of DeepMSA2 to provide more informative MSAs to a state-of-the-art protein prediction pipeline, thus further improving protein monomer modeling accuracy and rendering many previously 'un-foldable' proteins tractable for structure prediction.

7. The Reviewer commented:

8. Figure 4 convincingly shows that in spite of my reservations above DeepMSA2 managed to produce significantly better MSAs that enabled accurate prediction of nearly 2000 human

proteins for which AlphaFold models were not trustworthy. To further understand this apparent conflict, I explored a bit further. Indeed, the examples shown in the manuscript are very convincing. However, I clicked around in the database of ~5000 human proteins and in most cases the protein core, as predicted by DeepMSA2, seemed very similar to that of the AlphaFold model. I encourage the authors to highlight all the ~2000 models that they consider better than the corresponding AlphaFold models.

Thank you for raising the important point. We added a new **Figure S5** to examine the structural similarity between DMFold models and AlphaFold2 DB models on those 1,934 human proteins which AlphaFold2 failed to model but DMFold succeeded. As shown in **Figure S5**, 80% (=1549/1934) of the DMFold models have a different overall structure compared with the AlphaFold2 DB models (with TM-score <0.6 between the models), indicating that the improvement of the DMFold models for those proteins may come from the topology-level quality increasing. In contrast, for 385 targets, DMFold models have similar structures with AlphaFold2 DB model, where the improvement of DMFold model may come from the local topology correction.

Figure S5. Structural comparisons between DMFold models and AlphaFold2 DB models on 1,934 human proteins for which the DMFold creates high-quality models with pLDDT \geq 0.7, while AlphaFold2 DB models have a confidence score of pLDDT<0.7. The histogram shows the distribution of TM-scores between DMFold and AlphaFold2 DB on each target. There are 385 of 1,934 targets, where two methods generate similar modes with TM-scores between two methods' models of \geq 0.6, where the rest of 1549 (80%) have TM-score <0.6.

Accordingly, we have added the following paragraph to summarize the results of **Figure S5** (Page 5):

In Fig S5, we plot a histogram distribution of TM-scores between DMFold and AlphaFold2 DB models of the 1,934 proteins that could be folded only by DMFold. 80% (=1,549/1,934) of the DMFold models have a different overall structure relative to the corresponding AlphaFold2 DB models (with a TM-score <0.6 between them), indicating that the improvement of the DMFold models comes at the topology level. In contrast, for the remaining 385 proteins, DMFold models have relatively similar structures with AlphaFold2 DB models, and thus the improvements in DMFold may come mainly from local structural corrections.

8. The Reviewer commented:

9. Further on this: Having the per-residue measures that AlphaFold provide may have helped convincing me to the contrary.

This is an excellent point. To address this issue, we added **Figure 4B** to make a head-to-head comparison of the residue-level pLDDTs obtained by DMFold and AlphaFold2 DB respectively. In total, 878,094 residues are modeled by DMFold and AlphaFold2 DB for the 1,934 proteins, where DMFold models have greater pLDDTs than AlphaFold2 DB models for 93% of the residues. We added the following paragraph to summarize the results of **Figure 4B** (Page 6):

Fig 4B further shows a head-to-head comparison of the residue-level pLDDTs obtained by DMFold and AlphaFold2 DB for the 1,934 human proteins which involve a total of 878,094 residues, where the DMFold models have a higher residue-level pLDDT than the corresponding AlphaFold2 DB models on 93% of the residues.

Furthermore, we have given two examples and corresponding residue-level pLDDT distributions in **Figures 4C-4F** to illustrate the improvements of DMFold over AlphaFold2, which occur on both global fold and local structure levels. The corresponding discussions are given in the following paragraph (Page 6):

In **Fig 4C**, we present one illustrative example from an uncharacterized protein, Q6ZQT0, for which AlphaFold2 collects only 9 homologous sequences with $N_{eff}=0.7$ in the MSA, compared to the DeepMSA2 MSA with 122 sequences and $N_{eff}=6.2$. Due to the sparse information from the MSA, a poor structure model with pLDDT=0.51 is produced by AlphaFold2, showing an irregular secondary structure. In contrast, using the improved MSA from DeepMSA2, DMFold creates a model with much higher confidence (pLDDT=0.92), which has a more stable fold with well-formed hydrogen-bonding network and secondary structure. **Fig 4D** further lists the residue-level pLDDT distributions, where nearly all residues in the DMFold model have a pLDDT >0.7 , while the corresponding residues in the AlphaFold2 DB model all fell below 0.7. For this protein, the DMFold model and AlphaFold2 DB model have a very low similarity with TM-score=0.44, showing that DMFold improves the modeling quality at the global fold level. **Fig 4E** shows a complementary example from the putative diacylglycerol O-acyltransferase 2-like protein (Q6IED9) with a $\alpha\beta$ three-layer sandwich fold. Although the DMFold and AlphaFold2 models have similar global folds (TM-score=0.88), DMFold built the model with a pLDDT=0.83 while AlphaFold2 DB did so with a pLDDT=0.68. The residue-level pLDDT distributions in **Fig 4F** show that DMFold created better local structures with greater pLDDTs for several regions (marked in red), which correspond to the two better-formed β -sheets in 3D structural packing as highlighted in red in **Fig 4E**. These examples show that DMFold could improve AlphaFold2 modeling at both global fold and local structure levels through the supplying of additional evolutionary information from more informative MSAs.

Figure 4. The structural modeling results of DMFold on 5,042 difficult targets from the human proteome. (A) The histogram distributions of pLDDTs for DMFold models vs. AlphaFold2 DB models for the subset of 5,042 AlphaFold2 DB models with pLDDT < 0.70. The red dashed line marks the threshold pLDDT=0.70 for considering a target to be confidently predicted, where DMFold models have a pLDDT ≥ 0.7 in 1,934 cases. (B) A head-to-head comparison of the residue-level pLDDTs obtained by DMFold and AlphaFold2 DB for the 1,934 confidently modelled proteins which involve in total 878,094 residues. (C) Structural models generated for a putative uncharacterized protein FLJ45035 (Q6ZQT0) by AlphaFold2 DB (yellow) and DMFold (blue), respectively. (D) The residue-level pLDDT curves of AlphaFold2 DB (yellow) and DMFold (blue) for Q6ZQT0. (E) Structural models generated for a putative diacylglycerol O-acyltransferase 2-like protein (Q6IED9) by AlphaFold2 DB (yellow) and DMFold (blue), respectively, where two better-formed β -sheet secondary structures created by DMFold are highlighted by red. (F) The residue-level pLDDT curves of AlphaFold2 DB (yellow) and DMFold (blue) for Q6IED9, where the pLDDTs associated with the four β -stands are highlighted with red backgrounds.

9. The Reviewer commented:

10. I encourage the authors to include the per-residue pLDDT score in the database because it provides also local indications on structure quality, allowing to figure out which parts of the structure are more reliable.

Thank you for the suggestion. We have added residue-level pLDDT score figure in DMFold online database (see, e.g., each entry in the URL of <https://zhanggroup.org/DMFold2/human/1.html>).

10. The Reviewer commented:

11. And I am also missing the predicted aligned errors (PAE), i.e., the expected positional error at residue x if the predicted and actual structures are aligned on residue y.

Thank you for the suggestion. In this study, DMFold used an earlier released version of AlphaFold2 (v2.2.0) with pre-trained 'monomer' parameters to generate the human proteome models. Unfortunately, at that time, AlphaFold2 v2.2.0 does not have PAE output in the final 'pkl' file. And even with the newly released AlphaFold2 package (v2.3.0), it only has the "monomer_ptm" model supporting PAE calculations in the 'pkl'. In order to allow for the comparison requested by the Reviewer, we have rebuilt our human model database by re-modeling all of the 5,042 human proteins, using DMFold based on AlphaFold2 v2.3.0, which are now with both pLDDT and PAE values (see, e.g., <https://zhanggroup.org/DMFold2/human/Q6ZQT0/>).

In our benchmarking, we found that the models obtained based on AlphaFold2 v2.3.0 with the 'monomer_ptm' parameters have on average a slightly lower pLDDT than that on AlphaFold2 v2.2.0 with 'monomer' parameters (-this finding is consistent with the clarification by the DeepMind team that "This is the original CASP14 model fine tuned with the pTM head, providing a pairwise confidence measure. It is slightly less accurate than the normal monomer model", taken from <https://github.com/deepmind/alphafold>). Thus, in our manuscript, we still used models built on AlphaFold2 v2.2.0 'monomer' parameters for data analyses and discussions. But in the online database, we listed the models of both versions of DMFold programs, which allow users to choose different versions of models according to their needs (see, e.g., each entry in the URL of <https://zhanggroup.org/DMFold2/human/1.html>).

11. The Reviewer commented:

12. Figure 6. Since the structures are known it is trivial to superimpose them on the models. Thus, it makes sense to measure RMSD, which more accurately shows similarity.

Thank you for the suggestion. We have now added the RMSD values for the 27 complex targets in **Figure 6B**.

We previously listed TM-scores instead of RMSD in **Figure 6** partly because the CASP Assessors use the TM-score instead of RMSD as a part of their formula (Assessors' formula: $Z\text{-score(ICS)} + Z\text{-score(IPS)} + Z\text{-score(LDDTo)} + Z\text{-score(TM)}$) to evaluate the accuracy of protein complex structure models from different groups. Here, RMSD is the root mean squared deviation of all the equivalent atom pairs after the optimal superposition of the two structures. Because all atoms in the structures are equally weighted in the calculation, one drawback of RMSD is that big deviation on a few residues in the loop/rail regions could result in high RMSD values although the structures in the core regions are close, rendering the RMSD value more sensitive to the local structure deviations when RMSD value is big; this is especially an issue for protein complexes as they contain more loop and tail regions than monomers. In contrast, TM-score uses the Levitt-Gerstein factor which weights small distances stronger than big distances, which makes TM-score more

sensitive to the global topology. Thus, in the corrected manuscript, we listed both RMSD and TM-score in **Figure 6B**.

12. The Reviewer commented:

13. Methodology: Just to clarify, AlphaFold was retrained within the context of the improved MSA pipeline, right? Or maybe used as is?

Thank you for raising the question. We have chosen not to re-train AlphaFolds in both AlphaFold2 and DMFold pipelines for two reasons. First, we want to examine the impact of the DeepMSA2 on the structure prediction purely from the aspect of MSA improvement. Therefore, it is better to use the same model and parameters in both pipelines for fair comparisons. Second, because the original AlphaFold2 training sets include proteins with various MSA qualities with N_{eff} ranging from low to high values, we expect that the optimal weighting parameters of AlphaFold2 models should not change much when using the MSAs with an overall improved quality. In other words, retraining AlphaFold2 models should have a minimum impact on the final model quality with using new MSAs from DeepMSA2.

To clarify the issue, we have added the following paragraph in the “ONLINE METHODS” section (Page 18):

We note that we did not re-train the AlphaFold2 or AlphaFold2-Multimer network models with DeepMSA2 MSAs in the DMFold or DMFold-Multimer pipeline, since one focus of this study is on comparing the impact of the MSAs on protein structure prediction and making a fair comparison with AlphaFold2 and AlphaFold2-Multimer. Meanwhile, because the original AlphaFold2 training sets contain proteins with various MSA qualities –e.g., with N_{eff} ranging from low to high values–, it is expected that the retraining of AlphaFold2 models on a set of improved MSAs should have a minimal impact (if any) on the final model quality. Thus, for the calculations shown here, we simply used the DeepMind pre-trained AlphaFold2 and AlphaFold2-Multimer models and parameters when implementing DMFold and DMFold-Multimer programs.

13. The Reviewer commented:

14. When predicting the structure of a multimer, I take it that in the current implementation the stoichiometry is provided. I wonder however if it is possible to deduce it from the coevolution signal. In particular for homo-oligomers.

The Reviewer is right that the current implementation of the DMFold-Multimer pipeline requires the input of all protein component sequences, i.e., the providing of the stoichiometry information. Specifically, for homo-oligomers, the information of “ A_n ” with “ n ” being the number of identical sequences remains necessary.

We appreciate that the Reviewer raises a very interesting but challenging question of deducing the stoichiometry information from coevolution signals. First, we like to note that the stoichiometry state may not be unique for some proteins. For example, the human calcium homeostasis modulator can form an “A10” complex (PDB ID: 6YTV) and an “A11” complex (PDB ID: 6YTX), both of which are in stable form. Similarly, the proton-gated ion channel from *Gloeobacter*

violaceus can form both an “A5” homo-oligomer (PDB ID: 6zgj) and an “A6” homo-oligomer (PDB ID: 3igq). Another example is the portal protein from Escherichia phage T7, for which both “A12” and “A13” homo-oligomer states have been observed in the solved experimental structures (PDB IDs: 3j4a and 6qwp). Thus, solely based on a protein monomer sequence and its MSA, the co-evolution signal may not be sufficient to decide which oligomer states it should exist in.

Nevertheless, there are several machine learning studies trying to predict the protein oligomer states from query sequence alone, including, for example: (1) Shen, et. al., QuatIdent: A web server for identifying protein quaternary structural attribute by fusing functional domain and sequential evolution information. *Journal of Proteome Research*, 2009; (2) Y Sheng, et. al., Quad-PRE: a hybrid method to predict protein quaternary structure attributes. *Comput Math Methods Med*. 2014.

Accordingly, we have added the following paragraph in Section DISCUSSION to highlight the issue and the prospects of future developments, together with the citation of above efforts (Page 10):

In addition, the stoichiometry information (i.e., the number of copies of each component chain) of the complex is required before implementing the current DMFold-Multimer pipeline, which may limit the usefulness of the algorithm in practical applications. Including a deep learning-based stoichiometry predictor^{29, 30} based on the query sequences and evolutionary signals to DMFold-Multimer pipeline may be part of the solution to alleviating the limitation. Furthermore, whether the current DeepMSA2 approach could be extended to RNA MSA construction for improving RNA and RNA-protein complex structure prediction is also a topic to explore in our ongoing research.

Response to Reviewer #2

We very much appreciate the comments and suggestions from the Reviewer, which help to significantly improve the quality and description of the manuscript. One of the major concerns is on the validation of the human proteome structure prediction by DMFold. Accordingly, we have identified 48 newly solved protein structures and carefully examined the model accuracy relative to the experimental structures. Another major concern is on the lack of explanation why DMFold successfully modeled some of the immune proteins in CASP15 but not others. To address this issue, we have carefully analyzed the modeling data of two nanobody protein complexes H1142 and H1144 and found that the multiple MSA pairing and selecting procedure, which allows for the identification of correct MSAs with more relevant co-evolutionary information, is the key for the successful modeling of these immune protein structures by DMFold. In addition, the Reviewer pointed out a number of grammatical errors and points of unclear logic in the manuscript. We have carefully proofread the manuscript and partly rewritten the paragraphs/figures to address the grammar and presentation issues. We have also toned down several statements following the Reviewer's suggestion. Below, we include point-by-point replies to the comments of the Reviewer, where all changes have been highlighted in yellow in the manuscript.

1. The Reviewer commented:

Two main messages of the paper are that deeper and better constructed MSAs improve protein structure prediction and that this enhancement can push model accuracy beyond the 'vanilla' AlphaFold2 version (both standard and multimer), in many cases generating much better models. The method was tested in CASP15 and was recognized as one of the best performers. The study is well thought-through, the message is appropriately substantiated with the data, examples are illustrative and well picked, and the paper is overall well written.

We appreciate the positive comments of the Reviewer on the work.

2. The Reviewer commented:

Comments below.

ABSTRACT

'An integrated pipeline with DeepMSA2 participated in the most recent CASP15 experiment and created high quality complex models with mean Z-scores 3-times higher than the AlphaFold2-Multimer server.'

Even though technically correct, this statement may mislead readers to think that the reported method is 3 times better than AF2-Multimer. First and foremost, it is not an apple-to-apple comparison as the databases are different. I guess that if the available AF2-M method was simply retrained with the 'huge metagenomics data' (without any other changes), it would show comparable performance. This was demonstrated on monomeric CASP15 targets with an updated AF2 version 2.3.0 released in the end of 2022. I suggest the authors tone down this statement (here and in the Discussion).

We appreciate and agree the Reviewer's comments. Following the suggestion, we have rewritten the sentences in both ABSTRACT and DISCUSSION sections to tone down the statement and clarify the confusion:

An integrated pipeline with DeepMSA2 participated in the most recent CASP15 experiment and created complex structural models with considerable higher quality than the AlphaFold2-Multimer server.

[ABSTRACT, Page 1]

In the most recent community-wide blind test of CASP15, DMFold-Multimer achieved the highest modeling accuracy for complex structure prediction, with an average TM-score 15.4% and average ICS score 27.5% higher than the public March-2022 version of the AlphaFold2-Multimer server (registered as 'NBIS-AF2-multimer'), respectively, according to the assessor's criteria.

[DISCUSSION, Page 9]

3. The Reviewer commented:

INTRODUCTION

lines 57-59: These lines are formulated cautiously (which is good), however an unexperienced reader may still downplay a very important word 'some' in the phrase 'on some orphan sequences for which detectable homologous sequences do not exist'. While in the next paragraph the authors do emphasize that language models in CASP15 were in fact inferior to the state-of-the-art methods, it is not clear whether the methods mentioned here are among those discussed further on. As an easy fix, I would suggest substituting 'proved to be capable of generating' with 'can generate'.

62-65: Indeed, in CASP14, results in assembly prediction were not as excited as those in tertiary structure prediction. However, this changed in CASP15, and the authors are surely aware of it and should adjust their text accordingly.

Thank you for the appropriate suggestions. We have replaced the words 'proved to be capable of generating' with 'can generate' following the Reviewer's suggestion (Page 2):

Protein structure prediction methods that combine a protein language model with the AlphaFold2 end-to-end structure learning module, such as ESMFold¹¹ and OmegaFold¹², can generate better models than AlphaFold2 on some orphan sequences for which detectable homologous sequences do not exist.

We also rewrote the next paragraph to (1) update the changes of complex structural modeling in CASP15 and (2) mention specifically the performance of the language models in CASP15 include the above-mentioned methods (e.g., OmegaFold) (Page 2):

Despite the ongoing efforts of the field, the methods noted above do not significantly improve the overall prediction accuracy of monomer proteins. In addition, structure prediction for protein complexes remains an even more substantial challenge. In the CASP14 experiment, for example, satisfactory models (those with interface contact score >0.8) could only be built for 7% of tested protein complexes¹³, in CASP15, the situation was considerably improved, where the best-performance methods (including the pipeline introduced in this study) provided satisfactory models for up to 47% of cases¹⁴. However, there is still no evidence that the modifications made in most newly introduced AlphaFold2-based methods (e.g., ColabFold) can significantly improve the prediction performance for structural modeling of protein-protein complexes relative to AlphaFold2.

It is also notable that in the CASP15 experiment, the ostensibly MSA-free protein language models, such as OmegaFold, performed poorly on most of the targets with few homologous sequences¹⁵, suggesting that the lack of sufficient identifiable evolutionary information encoded in the protein language model is equivalently problematic to shallow MSAs for explicitly MSA-based methods.

4. The Reviewer commented:

METHODS

485-488: CASP assessment is usually done on domains (evaluation units), and not targets (some targets can contain domains of different homology-based categories). Please clarify that the numbers provided here (and in figures) are for domains. Also, typically CASP domain classification includes an overlap TBM/FM difficulty category. No mentioning of that here. Checking Table S9 I can see that TBM/FM domains were included as a subclass of a broader FM category. Please clarify in the text.

Thank you for pointing out the issue. Following the suggestion, we have replaced the word ‘targets’ by ‘domains’ for the CASP monomer structure prediction throughout the manuscript and Supplemental Information. We have also rewritten the first paragraph of “ONLINE METHODS” to explain our classification of domain types (Page 14):

Monomer proteins from CASP. 293 domains from monomer targets in the Critical Assessment of protein Structure Prediction (CASP) experiments were collected to benchmark the effect of DeepMSA2 on monomer protein structure prediction. The CASP experiments often classify the domains as TBM-easy, TBM-hard, TBM/FM and FM. To simplify the data analyses, we merged TBM-easy and TBM-hard domains as ‘template-based modeling (TBM) domains’, and TBM/FM and FM domains as ‘free modeling (FM) domains’ in this study. In our benchmarks, 48 FM domains and 64 TBM domains came from CASP13; 37 FM domains and 50 TBM domains were taken from CASP14; and 47 FM domains and 47 TBM domains were from CASP15 (Table S11).

5. The Reviewer commented:

486-487: ‘the effect of DeepMSA2 on monomer protein structure folding’. Protein folding does not depend on MSAs. It has its own rules and pathways, often unknown. However, it does seem like modern-day protein structure prediction depends on MSAs.

We thank the Reviewer for the clarification. We have changed the “protein structure folding” to “protein structure prediction” in the entire manuscript and Supplemental Information.

6. The Reviewer commented:

498-505: Do not understand arithmetic here. $13,838 + 6,757 = 20,595$. This is more than 20,389 proteins mentioned as the Human Proteome dataset in your paper (including the 2-residue peptide). Where does the discrepancy come from?

Thank you for the question, which helps us clarify the issue. The number 20,389 was from the UniProt human proteome record, while the number $13,838 + 6,757 = 20,595$ came from AlphaFold2 DB database. Since UniProt records are constantly changing based on the newly added data, the two numbers were slightly different. Given the variations in (and constant updating of)

the UniProt database, we believe it might be better to give an approximate number to avoid confusion. We have accordingly revised the paragraph as the following (Page 14):

Human proteome. The human proteome dataset contains more than 20,000 proteins or peptides with lengths between 2 and 34,350 AAs collected from UniProt (<https://www.uniprot.org/uniprotkb/?facets=reviewed%3Atrue&query=proteome%3AUP000005640>). In 2021, DeepMind released the AlphaFold2 model database, AlphaFold2 DB²⁴, which contains structure models predicted by AlphaFold2 for several reference proteomes, including the human proteome. However, only around 70% (13,838) of human proteins in AlphaFold2 DB have confident predictions with pLDDT \geq 0.7. From the remaining 6,757 proteins for which AlphaFold2 failed to create confident folds, we selected the 5,042 proteins with lengths <800 AAs for remodeling by DMFold.

7. The Reviewer commented:

507-513: There are many different Neff definitions and protocols to calculate them. It would be nice if the authors can help their readers to understand the Neff definition provided here. Practically all Neffs in the main text are discussed from the comparative analysis perspective, i.e., this method recovers more diverse sequences than the other, and thus is better. But what about the absolute values? What Neff values according to the provided definition indicate a diverse enough MSA and a useful for the prediction purposes evolutionary signal? 1, 2, 5, 10? Or in other words, how many diverse sequences (below 0.8 cutoff) are needed in a MSA for, say, 100- or 300-residue long protein? What about the coverage of the query sequence?

This is a very good question. To address the issue, we have added **Figure S10** and the following paragraph to explain in an intuitive way the concept of *Neff* and approximate thresholds of *Neff* and alignment coverage needed for deep learning-based structure prediction in ONLINE METHODS section (Pages 14-15):

Based on the definition in Eq. (1), MSAs with a more diverse set of sequence pairs with sequence identity <0.8 have the term $\sum_{m=1, m \neq n}^N I[S_{m,n} \geq 0.8]$ closer to 0, and thus result in a higher *Neff* value given the same number of sequences (*N*). In case that all *N* sequences in an MSA are diverse (pairwise sequence identity <0.8), the term $\sum_{m=1, m \neq n}^N I[S_{m,n} \geq 0.8] = 0$, and then *Neff* will be N/\sqrt{L} . In other word, given a *Neff* cutoff *Neff_{cut}*, the minimal number of diverse sequences needed for modeling can be roughly estimated by

$$N_{min} = Neff_{cut} * \sqrt{L} \quad (2)$$

It is generally believed that MSAs with more diverse sequences and higher alignment coverages can provide more evolutionary and co-evolutionary information and thus better assist deep learning protein structure prediction. To quantitatively evaluate that belief based on our data, we plot in **Fig S10A** the TM-score of DMFold models versus *Neff* values for the 62 monomer FM domains in CASP13-15. Although higher *Neffs* tend to correspond to models with higher TM-scores, there is no clear quantitative threshold of *Neff* corresponding to the absolute success of structure modeling. Following the general trend, we can provide two approximate thresholds, *Neff*=2⁰ and *Neff*=2⁴, which are roughly associated with three TM-score territories, i.e., the average TM-scores with *Neff* in [0, 2⁰], (2⁰, 2⁴], and (2⁴, ∞) are roughly <0.70, ~0.85, and >0.90, respectively. Thus, following Eq. (2), approximately at least 10 (=2⁰ * $\sqrt{100}$) or 160 (=2⁴ * $\sqrt{100}$) diverse sequences are required for good- or high-accuracy modeling of a 100-residue long protein, respectively.

In **Fig S10B**, we also present a comparison of TM-score vs alignment coverage, which is defined as the average rate of aligned residues on the query sequence cross all homologous sequences in the MSA, for the same set of recent CASP targets. The data shows no obvious correlation between TM-score and coverage of the MSA. It is obvious that an MSA with coverage that is too short (e.g., with

the alignment focused only on the N-terminal tail) is useless for deducing co-evolutionary signal of the entire protein sequence. Our data suggest, however, that the final performance of structure prediction does not depend on the alignment coverage as long as the alignment covers a reasonable region of the query sequence (e.g., $>60\%$ in our case).

Figure S10. TM-score of DMFold models versus (A) the N_{eff} of DeepMSA2 MSAs, and (B) the alignment coverage between the query and homologous sequences of the DeepMSA2 MSAs on 62 CASP13-15 'FM' monomer protein domains. The 'FM' domains that came from protein complex are excluded in this analysis due to possible interference from binding partners. The red line indicates the average TM-score in each N_{eff} bin. Two approximate thresholds, $N_{eff}=2^0$ and $N_{eff}=2^4$, are plotted by blue dashed lines. The average TM-scores with N_{eff} lower than 2^0 , between 2^0 and 2^4 , and higher than 2^4 are roughly below 0.70, approximate 0.85, and higher than 0.90, respectively. If a domain does not have any homologous sequence in the MSA, we define the coverage as 0.

8. The Reviewer commented:

Figure 6: recommend swapping names of panels (A <-> B) in the figure and the caption. It was confusing for the reviewer that the upper left panel in the Figure was B and not A.

Thank you for the suggestion. We have redesigned the **Figure 6** panels to make it logically clearer. Now, the panels A, B and C are in top, middle, and bottom regions of **Figure 6** respectively.

9. The Reviewer commented:

773-775: Do not like 3 things: 1) the 'DMFold-Multimer' typo, 2) text in the 1st parenthesis, and 3) the fact that it is not clarified that the AF2-Multimer results are from the March 2022 edition of a public server. Suggest correcting to: (B) Sum of Z-scores on 43 multimeric targets for 87 CASP15 assembly groups. DMFold-Multimer (registered as 'Zheng') and the public March-2022 version of the AlphaFold2-Multimer server (registered as 'NBIS-AF2-multimer') are marked in red and yellow, correspondingly.

Thank you for the suggestion. We have fixed the typo and revised the text in the legend of **Figure 6** to clarify the point (Page 24):

Figure 6. Performance of the DMFold/DMFold-Multimer pipeline for protein complex structure prediction in the CASP15 experiment. (A) Histogram of the TM-scores of structural models by DMFold-Multimer on the 38 complex targets that have experimental structure released. (B) The first models produced by DMFold-Multimer superposed on the experimental structures for the 27 complex targets with TM-scores >0.8 , where the component monomers of the predicted models are shown in distinct colors with the experimental structures marked in black. (C) Sum of Z-scores on 41 multimeric targets for the 87 registered CASP15 assembly groups, with data taken from the CASP15 webpage. DMFold-Multimer (registered as 'Zheng') and the public March-2022 version of the AlphaFold2-Multimer server (registered as 'NBIS-AF2-multimer') are marked in red and yellow, respectively.

Furthermore, we have changed the "AlphaFold2-Multimer" group name to "the public March-2022 version of the AlphaFold2-Multimer server" in CASP15-related analyses in RESULTS and DISCUSSION Sections to avoid further confusion:

Overall, DMFold-Multimer achieved a cumulative Z-score of 35.30, which is nearly 3 times higher than that of the 'NBIS-AF2-multimer' group (i.e., the public March-2022 version of the AlphaFold2-Multimer server run by the Elofsson Lab on CASP15 targets, which achieved a cumulative Z-score of 12.27), and 21.1% higher than the second-best performing group (29.15). [Page 8 in RESULTS]

In the most recent community-wide blind test of CASP15, DMFold-Multimer achieved the highest modeling accuracy for complex structure prediction, with an average TM-score 15.4% and average ICS score 27.5% higher than the public March-2022 version of the AlphaFold2-Multimer server (registered as 'NBIS-AF2-multimer'), respectively, according to the assessor's criteria. [Page 9 in DISCUSSION]

10. The Reviewer commented:

RESULTS

96-97: 'The optimal multimer MSAs are then selected based on a combined score of the depth of the MSAs and folding score of the monomer chains'. Did not see it in the Methods where the 'folding score' of monomer chains is defined or used.

Thank you for pointing out the issue. Here, ‘folding score’ refers to ‘pLDDT’ score, which is given in Eq. (3) in ONLINE METHODS. We have revised the text in the first paragraph of the RESULTS section to clarify the point (Page 3):

The optimal multimer MSAs are then selected based on a combined score of the depth of the MSAs and folding score (i.e., pLDDT score) of the monomer chains as defined in Eq. (3) below.

11. The Reviewer commented:

98: There is no ‘Methods’ section (line 482 lists ‘ONLINE METHODS’, adjust either here or there).

Thank you for picking up the error. We have replaced ‘Methods’ by ‘ONLINE METHODS’ throughout the manuscript.

12. The Reviewer commented:

133: Well, $TM=0.483$ is not considerably higher than 0.469 or 0.477. They are all rather very similar. So nothing to brag about here.

Thank you for the comment. In the revised manuscript, to have a better comparison, especially on the proteins outside the AlphaFold2 training datasets, we included 94 new domains from CASP15 into our test dataset (see Table S11). Thus, we have redone the corresponding experiments on the enlarged dataset and also calculated the p -values between DeepMSA2 and other methods. We found that the DeepMSA2 method has now an average TM-score of 0.492, compared to that by BLAST (0.454), MMseqs2 (0.469), HHblits (0.463), HMMER (0.448), or PSIBLAST (0.448), where the p -values by one-sided Student’s t -test are all below $2.72E-08$ in the comparisons, as shown in Table S2. Accordingly, we have updated the data and suitably qualified the statement in the RESULTS (Page 4):

The average TM-score for all 293 CASP domains obtained using the MSA from DeepMSA2 (0.492) is also higher than those using the MSAs from BLAST (0.454), MMseqs2 (0.469), HHblits (0.463), HMMER (0.448), or PSIBLAST (0.448), all with p -values $<2.72E-08$ by one-sided Student’s t -test (Table S2).

13. The Reviewer commented:

135, 140, Figure 2: What does the ‘top L long-range contacts’ mean? Is L defined anywhere that I missed? And the phrase ‘top $5L$ long-range distance map’ is simply confusing.

Thank you for the question, which help us clarify the issue. Here, L is the protein length (which was defined in Eq. 1 but not here), and “long-range” represents a separation of at least 24 residues in the primary sequence. Thus, ‘top L long-range contacts’ refer to the L long-range contacts that are top ranked by the predicted contact probability from the DeepPotential. Similarly, ‘top $5L$ long-range distance map’ refers to the $5*L$ long-range distance predictions as ranked by the predicted distance probability from the DeepPotential.

We have revised the corresponding text in “Results” section (Page 4) and in **Figure 2** caption (Page 21) to clarify the point.

In **Fig 2B**, we present the precision of the top L long-range contact predictions made by the deep neural-network program DeepPotential^{11, 23}, using co-evolutionary features derived from the six different MSAs (where L is the query sequence length, and “long-range” represents a sequence separation $|i - j| \geq 24$ residues for the contacts between residues i and j , which are then ranked by the DeepPotential contact probability). Again, utilizing the DeepMSA2 MSA results in a higher precision of top L long-range contacts ($=0.601$) predicted by DeepPotential, compared to those obtained while using the MSAs from BLAST ($=0.514$), MMseqs2 (0.568), HHblits (0.559), HMMER (0.538), or PSIBLAST (0.566) as inputs for DeepPotential. A similar tendency can be seen for top $L/5$ and $L/2$ predictions as detailed in **Table S3**, where p -values are below $1.23E-05$ for all the comparisons when all domains in our evaluation set are considered.

In **Fig 2C**, we further display the mean absolute distance error (MAE , see Eq. 6) of the top $5L$ long-range distances predicted by DeepPotential, where the use of the MSA from DeepMSA2 results in an $MAE=2.22\text{\AA}$, which is significantly lower than those from the other five MSA programs, i.e., 3.09\AA (p -value= $1.62E-22$) for BLAST, 2.70\AA (p -value= $5.79E-12$) for PSIBLAST, 2.68\AA (p -value= $1.03E-07$) for MMseqs2, 2.74\AA (p -value= $1.26E-09$) for HHblits, and 2.98\AA (p -value= $4.80E-13$) for HMMER (**Table S4**). In **Fig S3**, we also display a head-to-head comparison of the MAE between DeepMSA2 and the five control methods, where DeepMSA2 has lower MAE values than the other MSA methods for a dominant fraction of the domains; this accounts for the major reason for the significant p -values. Overall, these data show that the balanced and highly informative MSA construction provided by DeepMSA2 might have encoded more relevant co-evolutionary features and help guide accurate template recognition and spatial restraint predictions; this ability is also important for the subsequent deep learning-based tertiary structure prediction.

Figure 2. Comparisons of MSAs generated by DeepMSA2 and five control methods for assisting template recognition and deep-learning spatial restraint prediction on 293 CASP13-15 monomer domains. (A) the average TM-score of the first template detected by HHsearch; (B) the precision of top L long-range residue-residue contact prediction with L being the sequence length and sequence separation $|i,j| \geq 24$; and (C) the mean absolute error (MAE) for the top $5L$ long-range residue-residue distance predictions by DeepPotential. The CASP domains are categorized into Free-Modeling (FM) and template-based modeling (TBM) by the accessors. The height of the histogram indicates the mean value and error bar depicts the 95% confidence interval for each variable using Student's t -distribution.

14. **The Reviewer commented:**

138-139: *What are the numbers in parentheses defining accuracy of a contact map?*

Thank you for the question, which help us clarify the presentation of our data. The numbers in parentheses represent the precision of contacts predicted by DeepPotential with input MSAs from DeepMSA2, BLAST, MMseqs2, HHblits, HMMER, or PSIBLAST. We have rewritten the corresponding text in RESULTS section to clarify the point (Page 4):

Again, utilizing the DeepMSA2 MSA results in a higher precision of top L long-range contacts ($=0.601$) predicted by DeepPotential, compared to those obtained while using the MSAs from BLAST ($=0.514$), MMseqs2 (0.568), HHblits (0.559), HMMER (0.538), or PSIBLAST (0.566) as inputs for DeepPotential.

15. **The Reviewer commented:**

139: 'A similar tendency can be seen FOR short- and ...'

Thank you for the suggestion. We have changed “from” to “for” here. Also, because the short- and medium-range contacts are less important, we focused our discussion mainly on the long-range contact predictions with different cutoffs. The new text now reads as following (Page 4):

A similar tendency can be seen for top $L/5$ and $L/2$ predictions as detailed in Table S3, where p -values are below $1.23E-05$ for all the comparisons when all domains in our evaluation set are considered.

16. The Reviewer commented:

135-146: *When I look at graphs in Fig. 2 (in particular, E and F) I cannot see anything SIGNIFICANTLY better than the rest (as the authors continuously emphasize). In the graphs, BLAST is always provided second after the DeepMSA2 and it gives an impression of a sizeable improvement. But BLAST results are expected to be poorer (as BLAST is a conceptually different and simpler method), and if one removes BLAST's violin in the lineup of methods (or moves it to the very right), then the 'significantly better' impression disappears. It is better to move to a quantitative language in the text.*

Thank you for the suggestions. In order to improve the presentation of **Figure 2**, we made the following changes on it: (1) we moved the less relevant data on the MSA parameters (N_{eff} , $SeqID$ and Cov , see Point 7 above) to **SI** and had **Figure 2** now focused on the parameters (TM-score, contact/distance accuracy) that are more relevant to deep-learning structure prediction; (2) we redid the statistics based on an enlarged test dataset including CASP15 domains; (3) given that the violin plots require the display of minimum and maximum values which results in a very wide y-axis range, we converted the violin plots to barplots with 95% confidence intervals which better highlights the differences among different methods; and (4) following the Reviewer's suggestion, we calculated the P -value in one-sided Student's t-test between DeepMSA2 and other control methods which are all consistently and significantly below 0.05 in all the comparisons when All domains are considered, as shown in **Tables S2-4**.

Accordingly, we have rewritten the following paragraphs based on the new analysis (Pages 3-4):

Nevertheless, the parameters considered above (N_{eff} , $SeqId$, cov) only measure the geometrical characters of the MSA matrix, and do not necessarily reflect the inherent evolutionary and co-evolutionary information contained in the MSAs (**Fig S2**), which are critical to the deep learning-based protein structure predictions. As a more direct test of their ability to encode evolutionary and co-evolutionary information, we further examine the performance of the MSAs in assisting template recognition and deep learning spatial restraint prediction. In **Fig 2A**, we list the average TM-scores of the structure templates recognized by HHsearch based on the profile HMMs constructed from the six different MSAs, where all close homologous templates (>30% sequence identity to the query sequence) have been excluded from the template library. It is shown that the templates detected using the DeepMSA2 MSA have the highest TM-score for both FM and TBM domains. The average TM-score for all 293 CASP domains obtained using the MSA from DeepMSA2 (0.492) is also higher than those using the MSAs from BLAST (0.454), MMseqs2 (0.469), HHblits (0.463), HMMER (0.448), or PSIBLAST (0.448), all with p -values $< 2.72E-08$ by one-sided Student's t-test (**Table S2**).

In **Fig 2B**, we present the precision of the top L long-range contact predictions made by the deep neural-network program DeepPotential^{11, 23}, using co-evolutionary features derived from the six

different MSAs (where L is the query sequence length, and “long-range” represents a sequence separation $|i - j| \geq 24$ residues for the contacts between residues i and j , which are then ranked by the DeepPotential contact probability). Again, utilizing the DeepMSA2 MSA results in a higher precision of top L long-range contacts ($=0.601$) predicted by DeepPotential, compared to those obtained while using the MSAs from BLAST ($=0.514$), MMseqs2 (0.568), HHblits (0.559), HMMER (0.538), or PSIBLAST (0.566) as inputs for DeepPotential. A similar tendency can be seen for top $L/5$ and $L/2$ predictions as detailed in Table S3, where p -values are below $1.23E-05$ for all the comparisons when all domains in our evaluation set are considered.

In Fig 2C, we further display the mean absolute distance error (MAE , see Eq. 6) of the top $5L$ long-range distances predicted by DeepPotential, where the use of the MSA from DeepMSA2 results in an $MAE=2.22\text{\AA}$, which is significantly lower than those from the other five MSA programs, i.e., 3.09\AA (p -value= $1.62E-22$) for BLAST, 2.70\AA (p -value= $5.79E-12$) for PSIBLAST, 2.68\AA (p -value= $1.03E-07$) for MMseqs2, 2.74\AA (p -value= $1.26E-09$) for HHblits, and 2.98\AA (p -value= $4.80E-13$) for HMMER (Table S4). In Fig S3, we also display a head-to-head comparison of the MAE between DeepMSA2 and the five control methods, where DeepMSA2 has lower MAE values than the other MSA methods for a dominant fraction of the domains; this accounts for the major reason for the significant p -values. Overall, these data show that the balanced and highly informative MSA construction provided by DeepMSA2 might have encoded more relevant co-evolutionary features and help guide accurate template recognition and spatial restraint predictions; this ability is also important for the subsequent deep learning-based tertiary structure prediction.

Figure 2. Comparisons of MSAs generated by DeepMSA2 and five control methods for assisting template recognition and deep-learning spatial restraint prediction on 293 CASP13-15 monomer domains. (A) the average TM-score of the first template detected by HHsearch; (B) the precision of top L long-range residue-residue contact prediction with L being the sequence length and sequence separation $|i-j| \geq 24$; and (C) the mean absolute error (MAE) for the top $5L$ long-range residue-residue distance predictions by DeepPotential. The CASP domains are categorized into Free-

Modeling (FM) and template-based modeling (TBM) by the accessors. The height of the histogram indicates the mean value and error bar depicts the 95% confidence interval for each variable using Student's t-distribution.

17. The Reviewer commented:

152: There were not that many FM targets in CASP13 + CASP14. I guess that the 85 FM mentioned in the text is for a wider target set including slightly easier FM/TBM domains.

The reviewer is right that the 85 FM domains include both FM and FM/TBM domains, which was to increase the statistics of FM domains. In the revised manuscript, we have included 94 new domains from CASP15 to further increase the size of the test dataset. We have rewritten the first paragraph of "ONLINE METHODS" to explain the setting of test dataset (Page 14):

Monomer proteins from CASP. 293 domains from monomer targets in the Critical Assessment of protein Structure Prediction (CASP) experiments were collected to benchmark the effect of DeepMSA2 on monomer protein structure prediction. The CASP experiments often classify the domains as TBM-easy, TBM-hard, TBM/FM and FM. To simplify the data analyses, we merged TBM-easy and TBM-hard domains as 'template-based modeling (TBM) domains', and TBM/FM and FM domains as 'free modeling (FM) domains' in this study. In our benchmarks, 48 FM domains and 64 TBM domains came from CASP13; 37 FM domains and 50 TBM domains were taken from CASP14; and 47 FM domains and 47 TBM domains were from CASP15 (Table S11).

18. The Reviewer commented:

188-191: Please reformulate - hard to read.

We have rewritten the paragraph to improve the readability (Page 5):

In Fig S4, we further list eight other examples from CASP13-15 (T0991-D1, T1064-D1, T1125-D1, T1125-D2, T1125-D5, T1130-D1, T1169-D1, and T1169-D4), in which the TM-score improvements by DMFold are higher than 0.3. In seven out of these eight cases, the N_{eff} of DeepMSA2 is higher than that of AlphaFold2. These results again demonstrate the capacity of DeepMSA2 to provide more informative MSAs to a state-of-the-art protein prediction pipeline, thus further improving protein monomer modeling accuracy and rendering many previously 'un-foldable' proteins tractable for structure prediction.

19. The Reviewer commented:

198-199: have a confidence score (pLDDT<0.7) -> have a confidence score of pLDDT<0.7.

Thank you for the suggestion and we have made the correction.

20. The Reviewer commented:

204: Please explain also in the Figure 4 caption what is the '1,934' number shown there (it points in the y-axis direction, which shows pLDDT). Also, explain (in addition to the main text) that the '5042 difficult targets' mentioned there are those specifically selected by you where AF2's pLDDT

was <0.7 , as by quickly looking at the histogram it may seem that AF2 is unable to generate models in excess of 0.7. In general, I think that the panel A may be confusing for a reader, and I would recommend (but not insist) to delete the AF2 histogram as it delivers no useful message in itself in the discussed context.

Thank you for the suggestion. The “1,934” refers to the number of human proteins where DMFold creates models with $pLDDT \geq 0.7$, while AlphaFold2 DB models have $pLDDT < 0.7$. To be clearer, we have changed ‘1934’ to ‘1934 proteins’ in **Figure 4A**.

The “5,042” refers to the number of human proteins where the AlphaFold2 DB models have $pLDDT < 0.7$, so we run DMFold on these 5,042 proteins to see the performance of DMFold. We have made it clearer in the caption of **Figure 4A** by stating “for the subset of 5,042 AlphaFold2 DB models with $pLDDT < 0.70$.”

Finally, we elected to keep the Panel A as a graphic comparison of the AlphaFold2 DB and DMFold histograms that could help illustrate the difference in $pLDDT$ distributions. The **Figure 4** and the caption are now read as (Page 22):

Figure 4. The structural modeling results of DMFold on 5,042 difficult targets from the human proteome. (A) The histogram distributions of $pLDDT$ s for DMFold models vs. AlphaFold2 DB models for the subset of 5,042 AlphaFold2 DB models with $pLDDT < 0.70$. The red dashed line marks the threshold $pLDDT=0.70$ for considering a target to be confidently predicted, where DMFold models have a $pLDDT \geq 0.7$ in 1,934 cases. (B) A head-to-head comparison of the residue-level $pLDDT$ s obtained by DMFold and AlphaFold2 DB for the 1,934 confidently modelled proteins which involve in total 878,094 residues. (C) Structural models generated for a putative

uncharacterized protein FLJ45035 (Q6ZQT0) by AlphaFold2 DB (yellow) and DMFold (blue), respectively. (D) The residue-level pLDDT curves of AlphaFold2 DB (yellow) and DMFold (blue) for Q6ZQT0. (E) Structural models generated for a putative diacylglycerol O-acyltransferase 2-like protein (Q6IED9) by AlphaFold2 DB (yellow) and DMFold (blue), respectively, where two better-formed β -sheet secondary structures created by DMFold are highlighted by red. (F) The residue-level pLDDT curves of AlphaFold2 DB (yellow) and DMFold (blue) for Q6IED9, where the pLDDTs associated with the four β -stands are highlighted with red backgrounds.

21. The Reviewer commented:

195-210:

1) *Have any predicted structures from the human proteome been solved experimentally recently? Especially from the 5,042? It would be interesting to compare the results to known answers.*

2) *Also, it is nice that in 1,943 cases the pLDDT score jumped over 0.7. However, I am curious, in how many cases these are true positives (i.e., represent better predictive ability of DMFold) compared to false positives (e.g., the target is largely disordered and low AF2's pLDDT scores are genuinely indicating that). But I guess it is impossible to check this without the experimental structural data, or?*

This is a very good suggestion. Following the suggestion, we identified 48 human proteins that were recently solved and used them to make a comparison between DMFold and AlphaFold2 DB models relative to the experimental structures with results summarized in **Table S6**. Furthermore, we also added a new **Figure S6** to examine the correlation between TM-score and pLDDT of DMFold for those 48 proteins. We added the following paragraph to discuss the data (Page 6):

Out of the 5,042 human proteins for which no high-confidence AlphaFold2 structure was available, 48 have experimental structures that cover >80% of the sequence of the natural protein and were released in the PDB after the model training date of AlphaFold2 (May 1, 2018). For these 48 proteins, AlphaFold2 DB models achieve an average TM-score of 0.630, compared to the average TM-score achieved by DMFold (=0.679), providing a significant improvement with p -value=1.46E-04 by one-sided Student's t-test (see **Table S6**). **Fig S6** examines the correlation between TM-score and pLDDT of DMFold for those 48 proteins. Among all models with DMFold pLDDT ≥ 0.7 , 85% of the predictions could be considered as true positives, i.e., the model is predicted as foldable and is actually foldable with TM-score >0.5. There is also a quite high false omission rate (76%) if solely based on the 0.7 pLDDT score cutoff, suggesting that many of the models with a lower pLDDT might also possess correct folds. Overall, we note that despite the higher scores for the small set of recently crystallized human proteins, the absolute quality of the predicted human proteome models from DMFold still needs to be verified with more proteins when the experimentally solved structures are available in the future.

Figure S6. A head-to-head comparison between TM-score and pLDDT for final models by DMFold on 48 human proteome proteins that have recently solved experimental structures, testing the performance of pLDDT as a binary classifier for whether a model is correctly folded (using $\text{pLDDT} \geq 0.7$ as the model-based prediction, and TM-score between the model and the experimental structure > 0.5 as the ground truth). 'TP' means the number of true positive models, where DMFold models are predicted as foldable with a $\text{pLDDT} \geq 0.7$, and are also actually foldable with a TM-score ≥ 0.5 . 'FP' means the number of false positive models, where DMFold models are predicted as foldable with a $\text{pLDDT} \geq 0.7$, but are actually non-foldable with a TM-score < 0.5 . 'TN' means the number of true negative models, where DMFold models are predicted as non-foldable with a $\text{pLDDT} < 0.7$, and are also actually non-foldable with a TM-score < 0.5 . 'FN' means the number of false positive models, where DMFold models are predicted as non-foldable with a $\text{pLDDT} < 0.7$, but are actually foldable with a TM-score ≥ 0.5 .

22. The Reviewer commented:

296-297: content: 'where the former two measure the global fold quantity' [?] and grammar: and the latter two for assessing protein interface modeling quality of protein complexes.

Thank you for the suggestion. We have corrected the typo and grammar as following (Page 8):

It is observed that DMFold-Multimer outperformed all other groups in terms of the sum of Z-score, which was calculated by the CASP Assessors based on a combination of TM-score, LDDT, Interface Contact Score (ICS), and Interface Patch Score (IPS); where the former two (TM-score and LDDT) measure the global fold quality and the latter two (ICS and IPS) assess the protein interface modeling quality of protein complexes.

23. The Reviewer commented:

297-298: 'The standard version of AlphaFold2-Multimer also participated in the CASP15 with registered name of 'NBIS-AF2-multimer' as operated by the Elofsson lab.' Despite your explanations, it still sounds like DeepMind participated in CASP15, which is not true. I guess the sentence in question can be deleted and then you can say: Overall, DMFold-Multimer achieved a cumulative Z-score of 35.43, which is nearly 3 times higher than that of the 'NBIS-AF2-multimer' group (i.e., the public AlphaFold2-Multimer server run by the Elofsson Lab on CASP15 targets) (12.30), and 18.3% higher than the second-best performing group (29.95).

Thank you for the suggestion. We have revised the text accordingly (Page 8):

Overall, DMFold-Multimer achieved a cumulative Z-score of 35.30, which is nearly 3 times higher than that of the 'NBIS-AF2-multimer' group (i.e., the public March-2022 version of the AlphaFold2-Multimer server run by the Elofsson Lab on CASP15 targets, which achieved a cumulative Z-score of 12.27), and 21.1% higher than the second-best performing group (29.15).

24. The Reviewer commented:

299-300: Please make sure your numbers are updated to reflect those provided at the link in line 293.

Thank you for raising the point. The CASP15 official website was updated on May 3 according to the statement of on website (https://predictioncenter.org/casp15/zscores_multimer.cgi): **** The performance graph and table were updated on May 3, 2023 to exclude two targets with unreliable experimental stoichiometry data. Originally, results for 43 targets were assessed. This change had marginal effect on the cumulative Z-scores and did not affect the group ranking reported at the CASP15 meeting.*

Accordingly, we have updated the data in **Figure 6C** and **Table S9**, as well as the main text in Page 8:

Overall, DMFold-Multimer achieved a cumulative Z-score of 35.30, which is nearly 3 times higher than that of the 'NBIS-AF2-multimer' group (i.e., the public March-2022 version of the AlphaFold2-Multimer server run by the Elofsson Lab on CASP15 targets, which achieved a cumulative Z-score of 12.27), and 21.1% higher than the second-best performing group (29.15). Specifically, the TM-score, LDDT, ICS, and IPS of DMFold-Multimer are 0.830, 0.789, 0.598, and 0.641, which are 15.4%, 9.7%, 27.5%, and 19.1% higher than the public March-2022 version of the AlphaFold2-Multimer (0.719, 0.719, 0.469, and 0.538), respectively, all with a significant p -values < 0.05 by one-sided Student's t -tests (Table S10).

25. The Reviewer commented:

301-310: Talking about the immune complexes. Can you explain why these three were modeled very well, while others that are similar (e.g., H1142) were not?

Thank you for raising the excellent question. We added two figures (Figures S8 and S9) in SI and the following paragraphs to carefully examine the reason why DMFold-Multimer modelled well (and outperformed AlphaFold2-Multimer) on some of the antibody-antigen complexes (H1140, H1141, H1143, H1144) but not others (H1142) (Pages 8-9):

In Fig S8, we take Target H1144 as an example to further examine the possible reason of the successful modeling by DMFold-Multimer compared to AlphaFold2-Multimer on the nanobody-antigen complexes. First, Fig S8A shows a 3D scattering plot of between TM-score, predicted TM-score, and N_{eff} of paired MSAs for the structural decoys by DMFold-Multimer. Here, DMFold-Multimer implemented 25 paired MSAs created by DeepMSA2 which have N_{eff} s ranging from 1.8 to 16.3 and created 625 decoy conformations in which 13.6% have high quality with TM-score above 0.8. Importantly, there is a decent correlation between the actual TM-score and predicted TM-score in the high TM-score region (see the top-right area of the 2D TM-score vs predicted TM-score plane of Fig S8A), which allows for DMFold-Multimer to select a correct model with TM-score 0.99 based on the predicted TM-score. It is notable that this best model comes from the MSA with the highest N_{eff} (=16.3) that contains more abundant and relevant co-evolutionary information for quaternary structure modeling. As a comparison, Fig S8C displays the 3D scattering plot for the decoys by re-running AlphaFold2-Multimer, which utilized a single MSA with N_{eff} =8.1 and generated no models with TM-score above 0.8. In Figs S8B and 8D, we compare the distance map restraints for the models with the highest predicted TM-scores by DMFold-Multimer and AlphaFold2-Multimer, respectively. Although both programs have correct distance predictions for the intra-chain residues, only DMFold-Multimer has a correct distance map between the inter-chain residues (with a low MAE =0.61 Å, compared to 4.35 Å by AlphaFold2-Multimer), which is essential to the correct quaternary structure modeling. This example highlights the advantage of DMFold-Multimer by utilizing a multiple MSA pairing strategy to extract diverse co-evolutionary information which covers a more extensive quaternary conformational space, where the positive TM-score and predicted-TM-score correlation allows for the selection of correct complex models.

Nevertheless, DMFold-Multimer could not fold all the nanobody-antigen complexes in CASP15. In Fig S9, we present the modeling result on Target H1142, the only case out of the five nanobody-antigen targets in CASP15 for which DMFold-Multimer had the predicted model with a TM-score below 0.90 (-the TM-score of DMFold-Multimer model is 0.98 for H1143 which was not shown in Fig S7). Although DMFold-Multimer also uses multiple paired MSAs with N_{eff} ranging from 2.0 to 17.0 for this target, none of the MSAs contains correct inter-chain co-evolutionary information or created correct inter-chain distance maps as shown in Fig S9C. As a result, all the created conformational decoys have low quaternary TM-scores (Fig S9B), and the final model selected using predicted TM-score has thus a completely incorrect inter-chain orientation with a poor TM-score 0.614, despite the correct tertiary folding in the individual chains (Fig S9A). Interestingly, the maximum of predicted TM-scores for H1142 (pTM_{max} =0.75) is considerably lower than that of all other successfully folded nanobody-antigen complexes, i.e., pTM_{max} =0.82, 0.88, 0.90 and 0.90 for H1140, H1141, H1143, and H1144, respectively, suggesting that the pTM_{max} may be used as a potential indicator for estimating the quality of the DMFold-Multimer models in blind prediction experiments.

Figure S8. Case study of Target H1144 from the CASP15 Multimeric Modeling Section, which is a nanobody-antigen complex. (A) 3D scatter plot for TM-score, predicted TM-score, and N_{eff} of paired MSAs on DMFold-Multimer decoys. Here, the predicted TM-score is defined by $pTMS = 0.2 * pTM + 0.8 * ipTM$, where pTM and ipTM are predicted TM-scores for monomer and interface models, respectively, following AlphaFold2 modeling. The larger-sized cyan points are 3D points, representing DMFold-Multimer decoys with different TM-score, predicted TM-score, and N_{eff} of paired MSAs, where the red point refers to the 3D point corresponding to the decoy with the highest predicted TM-score. The smaller-sized black points represent the projection of 3D cyan points on the 2D planes, where the yellow points indicate the projection of the 3D red point on each of the 2D planes. Here, some of DMFold-Multimer decoys have very high TM-scores as well as high predicted TM-score, so they can be correctly selected as the final model based on the highest predicted TM-score. (B) The residue-residue distance map (heat map) for the model with the highest predicted TM-score from DMFold-Multimer (upper triangle) compared to that calculated from the experimental structure (lower triangle). (C) Same as in panel 'A', but modeled with AlphaFold2-Multimer. Note that the panel 'C' has the same number of points (decoys) as panel 'A', but most of points are overlapped, and no high-quality models are generated. (D) Same as in panel 'B', but modeled with AlphaFold2-Multimer.

Figure S9. Case study of target H1142 from the CASP15 Multimeric Modeling Section, which is a nanobody-antigen complex. (A) The experimental structure and the DMFold-Multimer model for H1142. (B) The 3D scatter plot for TM-score, predicted TM-score, and N_{eff} of paired MSAs on DMFold-Multimer decoys of H1142. The larger-sized cyan points are 3D points, representing DMFold-Multimer decoys with different TM-score, predicted TM-score, and N_{eff} of paired MSAs, where the red point is the 3D point corresponding to the decoy with the highest predicted TM-score. The smaller-sized black points represent the projection of 3D cyan points on each 2D plane, where the yellow points indicate the projection of the 3D red point on each of the 2D planes. (C) The residue-residue distance map (heat map) for the model with the highest predicted TM-score from DMFold-Multimer (upper triangle) versus that calculated from the experimental structure (lower triangle) for H1142.

26. The Reviewer commented:

A QUESTION related to the subject of the paper, but not directly to the material discussed in the paper:

The paper proves that deep learning methods can generate good structure models when evolutionary information is abundant (deep MSAs). How do the authors see perspectives of deep learning methods for the RNA structure prediction, where structural data are much sparser?

Thanks for raising the general and important question. In many aspects, deep learning (DL) based RNA structure prediction can be regarded as an analogy of DL-based protein structure prediction. However, the DL-based RNA structure prediction is currently hampered by two major factors. First, as the Reviewer pointed out, the DL models are mainly learned from nature structures, while the limited number of known RNA structures in the PDB is the major barrier for high-quality DL model training. While the development of novel network learning algorithms is important to learn robust structural patterns from limited structural data, integrating sub-optimized DL models with advanced physics- and knowledge-based approaches may be another feasible way to alleviate the difficulty. Although the statistical approaches have generally lower accuracy, they could provide complementary information to enhance accuracy and coverage of the DL models. The potential has been partly demonstrated in the RNA Structure Section of the CASP15 experiment, where statistical approaches generally outperformed the DL approaches. The outperformance is particularly significant for some of the computer-designed RNAs whose structural patterns are not contained in the limited RNA structure library, where the top statistical approaches assembled models with significantly higher TM-score than the DL models.

Secondly, the core information fed to DL networks is the evolutionary and coevolutionary information derived from homologous sequences of multiple sequence alignments. Most of the

current MSA collection programs are built on the NCBI non-redundant (NR) database, which contains only 96,714,726 RNA/DNA sequences in the 2023-08-01 version; this number is far smaller than that of proteins sequences in the NCBI NR database currently containing 595,907,626 sequences. Furthermore, as demonstrated in this study, the cutting-edge protein MSA algorithms take the advantage of metagenomic databases which are hundreds of times larger than the NCBI NR databases. Therefore, the extension of current DeepMSA2 to RNA MSA collections which enables the use of abundant metagenome RNA sequences is another urgent topic to explore in DL RNA structure prediction.

We have added the following paragraph to briefly note the potentials (Page 10):

Furthermore, whether the current DeepMSA2/DMFold approach could be extended to RNA and RNA-protein complex structure prediction is also a topic to explore in our ongoing research, where both limitations on the spare availability of RNA sequence and structure databases compared to proteins need to be overcome.

27. The Reviewer commented:

Noticed grammar issues: lines 95, 516-517, 538, 595, 645-646, 691-692, 185, 235-236

Thank you for picking up the grammar issues. We have carefully proofread the manuscript and corrected these and other issues.

Original line 95

Original:

Here, a set of M top ranked monomeric MSAs from each chain are paired those of all other chains, which result in M^m hybrid multimeric MSAs with m being the number of distinct monomer chains in the complex.

Revised:

Here, a set of M top ranked monomeric MSAs from each chain are paired **with** those of all other chains, which **results** in **M^N** hybrid multimeric MSAs with **N** being the number of distinct monomer chains in the complex.

Original line 516-517

Original:

Three genomic sequence databases, Uniclust30²¹, UniRef30²¹ and Uniref90¹⁸, which are all based UniProtKB³⁰, are utilized in the DeepMSA2 pipeline (see details in Table S10).

Revised:

Three genomic sequence databases, Uniclust30²², UniRef30²² and Uniref90¹⁹, which are all based **on** UniProtKB³¹, are utilized in the DeepMSA2 pipeline (see details in **Table S12**).

Original line 538

Original:

The 1.59 billion metagenomics sequence were clustered with 50% sequence identity at 90% coverage, resulting in 712 million clusters and the corresponding non-redundant sequences.

Revised:

The 1.59 billion metagenomics **sequences** were clustered with 50% sequence identity at 90% coverage, **yielding** 712 million clusters and the corresponding non-redundant sequences.

Original line 595

Original:

In addition, to speed up the custom database construction and filter out the noisy raw sequence picked up by Jackmmer and HHMsearch in Stages 2 and 3, respectively, a BLAST filter is adopted after the raw sequences picked up from Uniref90 before the kCust clustering.

Revised:

In addition, to speed up the custom database construction and filter out the noisy raw sequences picked up by Jackmmer and HHMsearch in Stages 2 and 3, respectively, a BLAST filter is applied to the raw sequences obtained from Uniref90 prior to kCust clustering.

Original line 645-646

Original:

Sequence linking. For a given set of $P = M^m$ paired monomeric MSAs, e.g. $(MSA_{l_1}, MSA_{l_2}, \dots, MSA_{l_P})$ with $1 \leq i \leq M$, the sequences from the monomeric MSAs are concatenated into a multimeric MSA as follows (see Fig S5B):

Revised:

Sequence linking. For a given set of M^N paired monomeric MSAs, e.g. $(MSA-1_{l_1}, MSA-2_{l_2}, \dots, MSA-N_{l_N})$ with $1 \leq l_1, l_2, \dots, l_N \leq M$, the sequences from the monomeric MSAs are concatenated into a multimeric MSA as follows (see Fig S12B):

Original line 691-692

Original:

Additional L2 regularization terms are also added to avoid possible over-fitting, where $\lambda_{single} = 1$ and $\lambda_{pair} = 0.2 \times (L - 1)$ are the regularization coefficients. The MI feature of residue i and j is defined by:

Revised:

Additional L2 regularization terms are also added to avoid possible over-fitting, where $\lambda_{single} = 1$ and $\lambda_{pair} = 0.2 \times (L - 1)$ are the regularization coefficients. The MI feature of residue pair i and j is defined by:

Original line 185

Original:

The better-quality model from DMFold is mainly due to the fact that DeepMSA2 constructs a deeper MSA with 42 homologous sequences and a Neff of 2.2 which offers more helpful co-evolutionary information.

Revised:

The improvement in modeling quality by DMFold is mainly because DeepMSA2 constructs a deeper MSA with 42 homologous sequences and a Neff of 2.2, which offers more helpful co-evolutionary information.

Original line 235-236

Original:

Compared with homomer complexes, the improvement of DMFold-Multimer over AlphaFold2-Multimer is relatively lower for heteromer complexes.

Revised:

Compared to homomer complexes, the magnitude of TM-score improvement of DMFold-Multimer over AlphaFold2-Multimer is relatively small for heteromer complexes.

Decision Letter, first revision:

Dear Dr. Zhang,

Thank you for submitting your revised manuscript "Improving deep learning protein monomer and complex structure prediction using DeepMSA2 with huge metagenomics data" (NMETH-A51909A). It has now been seen by the original referees and their comments are below. The reviewers find that the paper has improved in revision, and therefore we'll be happy in principle to publish it in Nature Methods, pending minor revisions to satisfy the referees' final requests and to comply with our editorial and formatting guidelines.

TRANSPARENT PEER REVIEW

Nature Methods offers a transparent peer review option for new original research manuscripts submitted from 17th February 2021. We encourage increased transparency in peer review by publishing the reviewer comments, author rebuttal letters and editorial decision letters if the authors agree. Such peer review material is made available as a supplementary peer review file. Please state in the cover letter 'I wish to participate in transparent peer review' if you want to opt in, or 'I do not wish to participate in transparent peer review' if you don't. Failure to state your preference will result in delays in accepting your manuscript for publication.

ORCID

Sincerely,
Arunima

Arunima Singh, Ph.D.
Senior Editor
Nature Methods

Reviewer #1 (Remarks to the Author):

I'm happy with how the authors revised the manuscript. The web-sites were down when I tried, but I'm sure that they are generally ok.
Congratulations!

Reviewer #2 (Remarks to the Author):

The authors significantly improved the paper, and I am happy with the most of responses. A few minor issues remain.

Response to #2.

Please polish the language in both paragraphs and add the version# of AF2-Multi in the Abstract.

#9, last paragraph.

"the public March-2022 version of the AlphaFold2-Multimer server
(registered as 'NBIS-AF2-multimer')"

still sounds like DeepMind enrolled it in CASP15. Add "run by the Elofsson Lab", like you did before.

#16

"the parameters considered above (Neff, SeqId, cov) only measure the geometrical characters of the MSA matrix"

Did not get this. What geometry characters?

#25

For the case study of immune complexes (H1140-44, Fig S8, S9), I do not think that paired MSAs (and the related Neffs) make sense as there is no evolutionary signal in inter-species complexes.

Author Rebuttal, first revision:

Response to Reviewer #1

1. The Reviewer commented:

Summary:

I'm happy with how the authors revised the manuscript. The web-sites were down when I tried, but I'm sure that they are generally ok.

Congratulations!

Thank you for the wonderful comments. We are pleased to know the Reviewer is satisfied with our revision. During the end of August to September 10, 2023, the internet of University of Michigan had an outage due to a university-wide security lockdown, and all the supercomputers of Yang Zhang Lab that are hosted in University of Michigan were temporarily shut down to the outside – this was a truly unprecedented situation for the University as a whole. We believe the Reviewer tried our websites at that time. All the Internet of our supercomputers has been recovered and DeepMSA2 and DMFold servers have been running correctly now. Thank you for the careful checking.

Response to Reviewer #2

We very much appreciate the comments and suggestions from the Reviewer, which helped us to further improve our manuscript. A main concern of the Reviewer is on the explanation of co-evolutionary information on the immune proteins. As described below, we have added further analyses to address the issue. The Reviewer also pointed out several language issues and unclear descriptions in the manuscript. We have carefully proofread the manuscript to fix the presentation issues. Below, we include point-by-point replies to the comments of the Reviewer, where all changes have been highlighted in yellow in the manuscript. Note that we had to move some contents to Supplementary Information (SI) because of the word limit by the Journal in the main manuscript. Nevertheless, we have ensured that the main text is self-contained and includes all major data and analyses supporting the conclusion of the manuscript.

1. The Reviewer commented:

The authors significantly improved the paper, and I am happy with the most of responses. A few minor issues remain.

We are pleased to know the Reviewer is satisfied with most of our revisions. We have included point-to-point responses to the Reviewer's comments below, which helped us in further strengthening the manuscript.

2. The Reviewer commented:

Please polish the language in both paragraphs and add the version# of AF2-Multi in the Abstract.

Thank you for the suggestion. We have proofread the entire manuscript and fixed a number of language issues. In addition, as suggested by the Reviewer, we added the AlphaFold2-Multimer version in the Abstract:

An integrated pipeline with DeepMSA2 participated in the most recent CASP15 experiment and created complex structural models with considerably higher quality than the AlphaFold2-Multimer server (v2.2.0).

3. The Reviewer commented:

*#9, last paragraph.
"the public March-2022 version of the AlphaFold2-Multimer server (registered as 'NBIS-AF2-multimer')" still sounds like DeepMind enrolled it in CASP15. Add "run by the Elofsson Lab", like you did before.*

We have added words in the third paragraph the Discussion section (Page 8):

In the most recent community-wide blind test of CASP15, DMFold-Multimer achieved the highest modeling accuracy for complex structure prediction, with an average TM-score 15.4% and average ICS score 27.5% higher than the public March-2022 version 2.2.0 of the AlphaFold2-Multimer

server run by the Elofsson Lab (registered as 'NBIS-AF2-multimer'), respectively, according to the assessor's criteria.

4. The Reviewer commented:

#16

"the parameters considered above (Neff, SeqId, cov) only measure the geometrical characters of the MSA matrix" Did not get this. What geometry characters?

Thank you for raising the question. By "geometry characters", we referred to the general feature characters collected from the sequence alignments themselves without additional training or decoding of the MSA matrix (such as training MSAs for contact and structure prediction and/or template recognition). We have removed the term of "geometry characters" and used instead more descriptive terminology. Please note that due to the space limitation of the main text, we moved the first section in original version of Results to Text S1 in SI:

Nevertheless, the parameters considered above (Neff, SeqId, cov) only measure general aspects of the alignment information of the MSAs, and do not necessarily reflect the inherent evolutionary and co-evolutionary information contained in the MSAs (Fig S3), which are critical to deep learning-based protein structure predictions.

5. The Reviewer commented:

#25

For the case study of immune complexes (H1140-44, Fig S8, S9), I do not think that paired MSAs (and the related Neffs) make sense as there is no evolutionary signal in inter-species complexes.

Thank you for the comment. We agree that the nanobody-antigen complex targets (H1140-H1144) have the component chains from different species and there may not be evolutionary signals in the inter-species complexes. However, this does not necessarily mean that paired MSAs are not important for nanobody-antigen complex structure predictions. In fact, it has been well established that deep-learning network models trained on the MSAs from homologous complex sequences could provide inter-chain structural patterns on how these component chains interact with each other. Such structural pattern information (e.g., inter-chain contact and distance maps) will provide useful and general help for protein complex structure prediction, no matter whether the target complexes have evolutionary signals or not, because proteins (including nanobody-antigen) form complex structures guided by general physical interacting laws, such as static electronic, van der Waals, hydrogen bonding, and hydrophobic interactions etc.

In our example of H1144, DeepMSA2 identified 413 paired sequences from the same species, which include, for example, two homologous sequences for the antigen are 2',3'-cyclic-nucleotide 3'-phosphodiesterases from *Homo sapiens*, and two homologous sequences for the query nanobody are Ig-like domains of *Homo sapiens*. These homologous proteins in the paired MSAs provide very useful information for the deep-learning networks to derive conserved structural patterns to help guide DMFold-Multimer to model the nanobody-antigen complex structures.

To clarify this point, we have added the following paragraph, as well as a new figure of **Figure S8C** which lists the top 100 ranked species based on the number of the paired sequences from the corresponding species in the paired MSA. Please note that due to the space limit, we have moved some of the related discussion on antibody-antigen into **Text S2** in SI:

Here, we note that although the nanobody and the antigen came from different species (i.e., alpaca and mouse) and there may not be evolutionary signals in the inter-species complexes, clear co-evolutionary signal could still be obtained in the DeepMSA2 paired MSAs, since the paired sequences of component MSAs are selected from the same species, which could be used to assist nanobody-antigen complex structure predictions (see “DeepMSA2-Multimer pipeline for multimeric MSA construction” section of the **Methods** in the main text). Indeed, we found that even though the nanobody itself is the product of adaptive immune molecule maturation, the DeepMSA2 MSAs provided information on how Ig-like folds can interact with folds resembling the target protein in other species, contributing to the quality of the resulting model. For the case of H1144, DeepMSA2 identified 413 paired homologous sequences that came from 172 common species (**Fig S8C**), where the co-evolution information contained in the paired sequences helps the deep learning networks learn the inter-chain distance restraints, resulting in an accurate predicted distance map (**Fig S8B**).

Figure S8. Case study of Target H1144 from the CASP15 Multimeric Modeling Section, which is a nanobody-antigen complex. (A) 3D scatter plot for TM-score, predicted TM-score, and N_{eff} of paired MSAs on DMFold-Multimer decoys. Here, the predicted TM-score is defined by $pTMS = 0.2 \cdot pTM + 0.8 \cdot ipTM$, where pTM and ipTM are predicted TM-scores for monomer and interface models, respectively, following AlphaFold2 modeling. The larger-sized cyan points are 3D points, representing DMFold-Multimer decoys with different TM-scores, predicted TM-scores, and N_{eff} of paired MSAs, where the red point refers to the 3D point corresponding to the decoy with the

highest predicted TM-score. The smaller-sized black points represent the projection of 3D cyan points on the 2D planes, where the yellow points indicate the projection of the 3D red point on each of the 2D planes. Here, some DMFold-Multimer decoys have very high TM-scores as well as high predicted TM-scores, so they can be correctly selected as the final model based on the highest predicted TM-score. (B) The residue-residue distance map (heat map) for the model with the highest predicted TM-score from DMFold-Multimer (upper triangle) compared to that calculated from the experimental structure (lower triangle). (C) Top 100 species contributing to the paired MSA for H1144 ranked by the number of paired sequences. (D) Same as in panel 'A', but modeled with AlphaFold2-Multimer. Note that the panel 'D' has the same number of points (decoys) as panel 'A', but most of points overlap, and no high-quality models are generated. (E) Same as in panel 'B', but modeled with AlphaFold2-Multimer.

Final Decision Letter:

Dear Dr. Zhang,

I am pleased to inform you that your Article, "Improving deep learning protein monomer and complex structure prediction using DeepMSA2 with huge metagenomics data", has now been accepted for publication in Nature Methods. Your paper is tentatively scheduled for publication in our January print issue, and will be published online prior to that. The received and accepted dates will be March 4, 2023 and November 13, 2023. This note is intended to let you know what to expect from us over the next month or so, and to let you know where to address any further questions.

Over the next few weeks, your paper will be copyedited to ensure that it conforms to Nature Methods style. Once your paper is typeset, you will receive an email with a link to choose the appropriate publishing options for your paper and our Author Services team will be in touch regarding any additional information that may be required.

You will receive a link to your electronic proof via email with a request to make any corrections within 48 hours. If, when you receive your proof, you cannot meet this deadline, please inform us at rjsproduction@springernature.com immediately.

Please note that *Nature Methods* is a Transformative Journal (TJ). Authors may publish their research with us through the traditional subscription access route or make their paper immediately open access through payment of an article-processing charge (APC). Authors will not be required to make a final decision about access to their article until it has been accepted. [Find out more about Transformative Journals](https://www.springernature.com/gp/open-research/transformative-journals)

Authors may need to take specific actions to achieve [compliance](https://www.springernature.com/gp/open-research/funding/policy-compliance-faqs) with funder and institutional open access mandates. If your research is supported by a funder that requires immediate open access (e.g. according to [Plan S principles](https://www.springernature.com/gp/open-research/plan-s-compliance)) then you should select the gold OA route, and we will direct you to the compliant route where possible.

For authors selecting the subscription publication route, the journal's standard licensing terms will need to be accepted, including [self-archiving policies](https://www.springernature.com/gp/open-research/policies/journal-policies). Those licensing terms will supersede any other terms that the author or any third party may assert apply to any version of the manuscript.

Your paper will now be copyedited to ensure that it conforms to Nature Methods style. Once proofs are generated, they will be sent to you electronically and you will be asked to send a corrected version within 24 hours. It is extremely important that you let us know now whether you will be difficult to contact over the next month. If this is the case, we ask that you send us the contact information (email, phone and fax) of someone who will be able to check the proofs and deal with any last-minute problems.

If, when you receive your proof, you cannot meet the deadline, please inform us at rjsproduction@springernature.com immediately.

Once your manuscript is typeset and you have completed the appropriate grant of rights, you will receive a link to your electronic proof via email with a request to make any corrections within 48 hours. If, when you receive your proof, you cannot meet this deadline, please inform us at rjsproduction@springernature.com immediately.

Once your paper has been scheduled for online publication, the Nature press office will be in touch to confirm the details.

Once your paper has been scheduled for online publication, the Nature press office will be in touch to confirm the details.

Content is published online weekly on Mondays and Thursdays, and the embargo is set at 16:00 London time (GMT)/11:00 am US Eastern time (EST) on the day of publication. If you need to know the exact publication date or when the news embargo will be lifted, please contact our press office after you have submitted your proof corrections. Now is the time to inform your Public Relations or Press Office about your paper, as they might be interested in promoting its publication. This will allow them time to

prepare an accurate and satisfactory press release. Include your manuscript tracking number NMETH-A51909B and the name of the journal, which they will need when they contact our office.

About one week before your paper is published online, we shall be distributing a press release to news organizations worldwide, which may include details of your work. We are happy for your institution or funding agency to prepare its own press release, but it must mention the embargo date and Nature Methods. Our Press Office will contact you closer to the time of publication, but if you or your Press Office have any inquiries in the meantime, please contact press@nature.com.

Nature Portfolio journals [encourage authors to share their step-by-step experimental protocols](https://www.nature.com/nature-research/editorial-policies/reporting-standards#protocols) on a protocol sharing platform of their choice. Nature Portfolio 's Protocol Exchange is a free-to-use and open resource for protocols; protocols deposited in Protocol Exchange are citable and can be linked from the published article. More details can found at www.nature.com/protocolexchange/about.

Best regards,
Arunima

Arunima Singh, Ph.D.
Senior Editor